# Genome-wide functional analysis reveals key roles for kinesins in the mammalian and mosquito stages of the malaria parasite life cycle

Mohammad Zeeshan[1¤]*, Ravish Rashpa[2☯], David J. P. Ferguson[3,4☯], Steven Abel[5☯], Zeinab Chahine[5], Declan Brady[1], Sue Vaughan[3], Carolyn A. Moores[6], Karine G. Le Roch[5], Mathieu Brochet[2], Anthony A. Holder[7], Rita Tewari [1]*

**1** University of Nottingham, School of Life Sciences, Nottingham, United Kingdom, **2** University of Geneva, Faculty of Medicine, Geneva, Switzerland, **3** Oxford Brookes University, Department of Biological and Medical Sciences, Oxford, United Kingdom, **4** University of Oxford, John Radcliffe Hospital, Nuffield Department of Clinical Laboratory Science, Oxford, United Kingdom, **5** Department of Molecular, Cell and Systems Biology, University of California Riverside, Riverside, California, United States of America, **6** Institute of Structural and Molecular Biology, Department of Biological Sciences, Birkbeck College, London, United Kingdom, **7** The Francis Crick Institute, Malaria Parasitology Laboratory, London, United Kingdom

☯ These authors contributed equally to this work.
¤ Current address: Faculty of Infectious and Tropical Diseases, London School of Hygiene and Tropical Medicine, London, United Kingdom.
* zeeshanmf@gmail.com (MZ); rita.tewari@nottingham.ac.uk (RT)

**Data Availability Statement:** Sequence reads for ChIP-seq and RNA-seq have been deposited in the NCBI Sequence Read Archive with accession

## Abstract

Kinesins are microtubule (MT)-based motors important in cell division, motility, polarity, and intracellular transport in many eukaryotes. However, they are poorly studied in the divergent eukaryotic pathogens *Plasmodium* spp., the causative agents of malaria, which manifest atypical aspects of cell division and plasticity of morphology throughout the life cycle in both mammalian and mosquito hosts. Here, we describe a genome-wide screen of *Plasmodium* kinesins, revealing diverse subcellular locations and functions in spindle assembly, axoneme formation, and cell morphology. Surprisingly, only kinesin-13 is essential for growth in the mammalian host while the other 8 kinesins are required during the proliferative and invasive stages of parasite transmission through the mosquito vector. In-depth analyses of kinesin-13 and kinesin-20 revealed functions in MT dynamics during apical cell polarity formation, spindle assembly, and axoneme biogenesis. These findings help us to understand the importance of MT motors and may be exploited to discover new therapeutic interventions against malaria.

## Introduction

Kinesins are microtubule (MT)-based motor proteins that use energy from the hydrolysis of ATP and function in various cellular processes including intracellular transport, mitotic spindle formation and chromosome segregation during cell division, and the organisation of cell

number: PRJNA731497. All other data are available in the main text or the supplementary materials.

**Funding:** This work was supported by: MRC UK (G0900109, G0900278, MR/K011782/1) and BBSRC (BB/N017609/1) to RT; the BBSRC (BB/N017609/1) to MZ; the Francis Crick Institute (FC001097), the Cancer Research UK (FC001097), the UK Medical Research Council (FC001097), and the Wellcome Trust (FC001097) to AAH; the NIH/NIAID (R01 AI136511) and the University of California, Riverside (NIFA-Hatch-225935) to KGLR; the Swiss National Science Foundation project grant (31003A_179321) to MB and BBSRC (BB/N018176/1) to CAM. This research was funded in whole, or in part, by the Wellcome Trust [FC001097]. The funders had no role in study design, data collection and analysis, decision to publish, or preparation of the manuscript.

**Competing interests:** The authors have declared that no competing interests exist.

**Abbreviations:** AID, auxin-inducible degron; dpi, days post-infection; GCβ, guanylate cyclase beta; GFP, green fluorescent protein; IFT, intraflagellar transport; IGV, Integrative Genomic Viewer; IMC, inner membrane complex; IP, intraperitoneally; ISP, IMC subcompartment protein; mAb, monoclonal antibody; mpa, min post-activation; MT, microtubule; PBS, phosphate buffered saline; PFA, paraformaldehyde; PTD, promotor trap double homologous recombination; qRT-PCR, quantitative real-time PCR; RBC, red blood cell; RT, room temperature.

polarity and cytoskeletal features associated with motility [1,2]. In eukaryotes, there are 14 to 16 kinesin subfamilies categorised according to the primary sequences of the motor domain, with similar biological roles also established by *in vitro* studies, and in vivo phenotypes for sub-family members [2–4]. Kinesin subfamilies that regulate MT dynamics, such as kinesin-8 and kinesin-13, are found in most eukaryotes including primitive and evolutionarily divergent eukaryotes [5,6]. Although there is an extensive kinesin literature with various bioinformatic and molecular investigations, information is sparse on these molecular motors in deep rooted pathogenic eukaryotes including *Plasmodium* spp. and other Apicomplexa, *Giardia* spp., and trypanosomes [6]. These primitive eukaryotes have a flagellate stage in their life cycle and may have a complex MT-associated cytoskeleton [7], indicating the importance of MT-based motor proteins in their development.

*Plasmodium* spp., the causative agents of malaria, belong to the phylum Apicomplexa. They are ancient haploid unicellular eukaryotes with several morphologically diverse proliferative stages during the complex life cycle in various cells, tissues, and organs of their vertebrate and invertebrate hosts (**Fig 1A**) [8,9]. In the mammalian host, the parasite proliferates within liver and red blood cells (RBCs) by repeated cycles of closed mitotic division retaining an intact nuclear membrane, with cytokinesis following the final nuclear division, in a process termed schizogony, to produce multiple infective haploid merozoites [10] (**Fig 1A**). Some of these haploid parasites in the RBC arrest and commit to sexual development as gametocytes (**Fig 1A**). Gametocytes develop no further into gametes unless ingested in a blood meal by a female mosquito (the invertebrate host). Male gametogenesis is very rapid and complete within 12 to 15 min after activation [11,12]. Within the nucleus, 3 rounds of DNA replication and chromosome segregation produce an 8N genome, which is followed by nuclear division and cytokinesis. At the same time, in the cytoplasm, axoneme assembly and maturation occur, leading to the formation of flagellate haploid male gametes in a process termed exflagellation [8,9]. The motile male gamete finds and fertilises the female gamete, and the resultant zygote differentiates through 6 distinct stages (I to VI) into a banana-shaped, invasive motile ookinete with a distinct apical polarity and conoid-associated proteins [8,13,14]. At the same time, in the first stage of meiosis, the DNA is duplicated and the now tetraploid ookinete develops over a 24-h period in the mosquito gut [8,14], before traversing the mosquito gut wall and forming an oocyst under the basal lamina. Within the oocyst, sporogony, which is a form of endomitosis, produces many haploid sporozoites [15,16]. Sporozoites are motile and invasive polarised cells that migrate to and invade the salivary glands, so that the bite of the infected mosquito injects them into the next mammalian host [17]. Overall, the complete life cycle of the malaria parasite is characterised by varied morphological differences in size and shape, together with various modes of cell division and proliferation (**Fig 1A**).

In a recent bioinformatic analysis of kinesins in Apicomplexa, we found 9 kinesins encoded in the *Plasmodium berghei* genome, with members of 3 conserved kinesin subfamilies (kinesin-5, kinesin-8B, kinesin-8X, and kinesin-13); kinesin-4, kinesin-15, and kinesin-20; and 2 Apicomplexa-enriched kinesins: kinesin-X3 and kinesin-X4 [18]. Surprisingly, kinesin-5, kinesin-8X, and kinesin-8B were not essential for blood stage proliferation [18–20]. However, deletion of *kinesin-5*, which codes for a protein clearly colocated with the spindle apparatus in all proliferative stages, affected the production of infective sporozoites [19]. Kinesin-8X was required for endomitotic proliferation in oocysts, and *kinesin*-8B deletion resulted in a defect in axoneme biogenesis during male gametogenesis [18,20,21].

Here, we present a comprehensive genome-wide screen of all *P. berghei* kinesins, including additional analyses of previously studied kinesin-5, kinesin,-8B, and kinesin-8X [18–20], using gene-targeting approaches, live cell imaging, ultrastructure expansion microscopy and electron microscopy, and RNA-seq and ChIP-seq analyses. We examine the subcellular location of

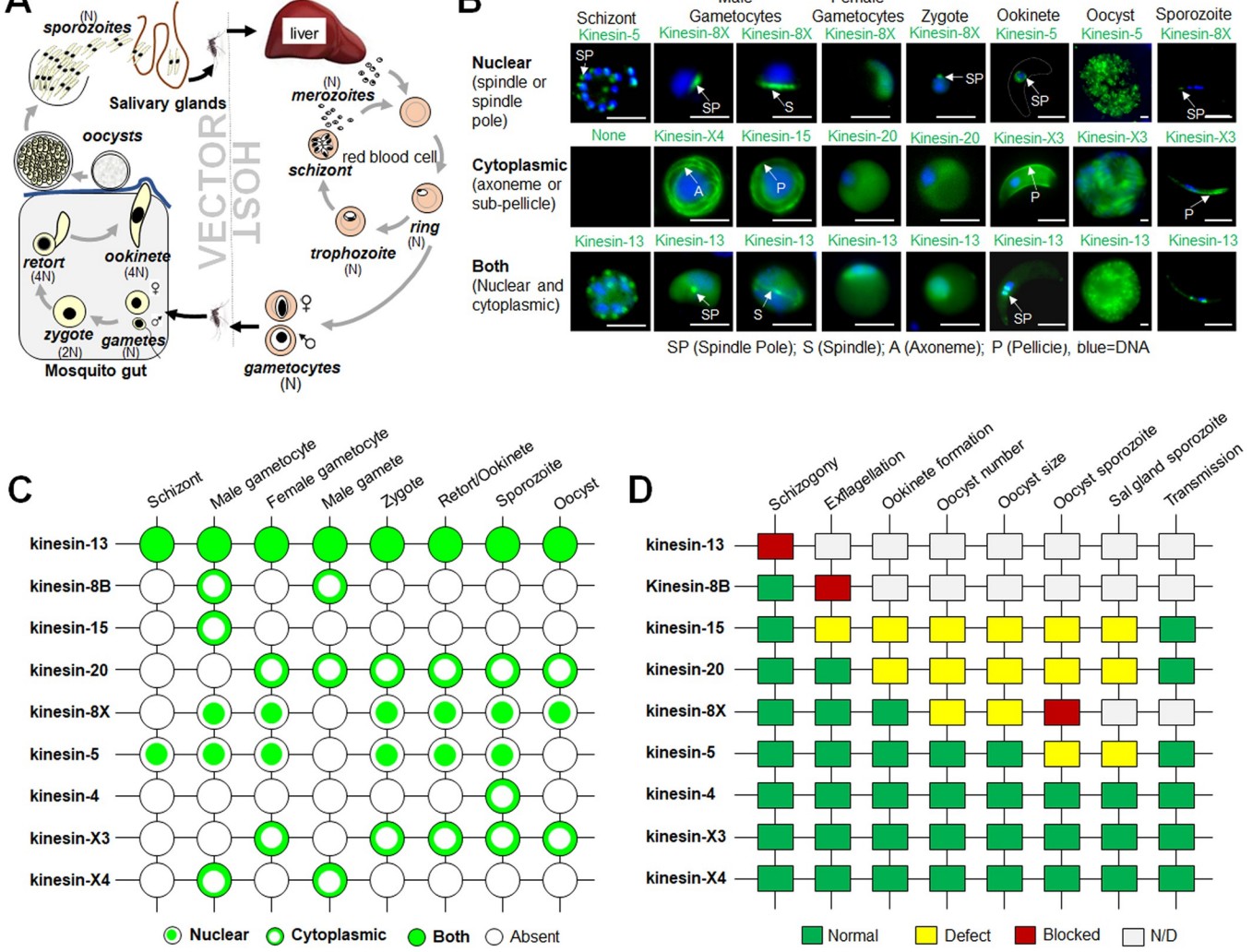

**Fig 1. Subcellular location and function of kinesins at various stages of the *Plasmodium berghei* life cycle.** (A) Life cycle of *Plasmodium* spp. showing different proliferative and invasive stages in host and vector. (B) Live cell imaging showing subcellular locations of representative kinesin-GFP proteins (green) during various stages of the *P. berghei* life cycle. DNA is stained with Hoechst dye (blue). Scale bar = 5 μm. (C) A summary of expression and location of kinesins-GFP during different stages of *P. berghei* life cycle. (D) Summary of phenotypes resulting from deletion of different kinesin genes at various stages of life cycle. The phenotype was examined for asexual blood stage development (schizogony), exflagellation (male gamete formation), ookinete formation, oocyst number, oocyst size, sporozoite formation in oocyst, presence of salivary gland sporozoites, and sporozoite transmission to vertebrate host. N/D, not determined.

each kinesin using a protein endogenously tagged at the C-terminus with GFP, revealing a differential localisation of kinesins in mitotic and meiotic stages and a pellicular and polar location in certain invasive stages. Eight of the 9 kinesin genes are required only for parasite transmission through the mosquito vector, during the sexual and sporogony stages. Only kinesin-13 is likely essential during blood stage schizogony. An in-depth analysis of kinesin-13 and kinesin-20 during gametocyte and ookinete stages revealed distinct subcellular locations and functions in MT spindle assembly and formation, axoneme assembly, and cell polarity. Kinesin-20 was associated with a striking ring-like structure during zygote to ookinete differentiation and deletion of the kinesin-20 gene revealed a function in the morphology and motility of the ookinete. Kinesin-13 is expressed at all proliferative stages of the life cycle, and it associates with the kinetochore. A kinesin-13 genetic knockdown affected MT dynamics during spindle

formation and axoneme assembly in male gametocytes, and subpellicular MT organisation in ookinetes. These findings help us understand the importance of MT motors and may be exploited to discover new therapeutic interventions against malaria.

## Results

### Live cell imaging of *Plasmodium* kinesins reveals diverse locations during cell division, differentiation, and pellicle formation throughout the life cycle

To investigate the expression and subcellular location of kinesins throughout the *P. berghei* life cycle, we generated transgenic parasite lines by single crossover recombination at the 3′ end of each gene to express a fusion protein with a C-terminal GFP-tag (**S1A Fig**). PCR analysis of genomic DNA from each line, using locus-specific diagnostic primers, indicated correct integration of the GFP sequence (**S1B Fig**). Immunoprecipitation assays using GFP-trap beads and mass spectrometry analysis of at least 6 kinesin-GFP proteins confirmed the presence of peptides of intact fusion protein in gametocyte lysates (**S2A and S2B Fig**). Each kinesin-GFP parasite line completed the full life cycle with no detectable phenotypic change resulting from the GFP tagging. We analysed the expression and subcellular location of these GFP-tagged proteins by live cell imaging at various stages of the life cycle. Taken together with the previously published results for kinesin-5, kinesin-8B, and kinesin-8X [18–20], we found that the 9 kinesins have a diverse pattern of expression, with distinct subcellular locations including the mitotic spindle, axonemes, the surface pellicle, and a polar distribution at various stages of the parasite life cycle (**Fig 1B and 1C**). Interestingly, only 2 kinesins, kinesin-5 and kinesin-13, were expressed throughout the parasite life cycle, including blood stage schizogony, and were located on the mitotic spindle in both asexual and sexual stages (**S3 Fig**). Kinesin-5GFP was restricted to the nucleus, while kinesin-13GFP had both a nuclear and cytoplasmic location (**S3 Fig**). Kinesin-8XGFP was also located on the nuclear spindle but only during the proliferative stages within the mosquito vector. Three kinesins (kinesin-8B, kinesin-15, and kinesin-X4) were expressed only during male gametogenesis with cytosolic locations (**S3 Fig**), and 2 kinesins (kinesin-20 and kinesin-X3) were first detected in female gametocytes with a diffuse location (**S3 Fig**). Their presence continues into the zygote and later stages of ookinete differentiation and sporogony with locations that are discussed in detail below. We also observed 2 kinesins at polar locations: kinesin-8X at the basal end of stage V to VI ookinetes and kinesin-13 at the apical end throughout ookinete development (**S3 Fig**). Overall, kinesin-5 and kinesin-8X are restricted to nuclear spindle and kinesin-13 is present in both nucleus and cytoplasm (**Fig 1B and 1C**). The Apicomplexa-enriched kinesin-X3 and kinesin-X4 are confined to ookinete and sporozoite pellicle and flagellar axoneme, respectively.

### Genome-wide functional screen reveals that 8 out of 9 kinesins are required only for parasite transmission and not for blood stage proliferation

Previously, we described the functional roles during mosquito stages of 3 kinesins, kinesin-5, kinesin-8B, and kinesin-8X, proteins that were not essential during blood stage development [18–20]. To study the function of the remaining 6 kinesins throughout the life cycle, we attempted deletion of each gene from *P. berghei* using a double crossover homologous recombination strategy as described previously [22] (**S4A Fig**). Successful integration of the targeting constructs at each gene locus was confirmed by diagnostic PCR (**S4B Fig**), except that *kinesin-13* could not be deleted. PCR analysis of knockout parasites confirmed the complete deletion of these *kinesin* genes (**S4B Fig**), indicating that they are not essential during the asexual blood

stage. *kinesin-13*, which could not be deleted despite several attempts, likely has an essential role during the asexual blood stage (**Fig 1D**). A recent functional profiling of the *P. berghei* genome [23] also supports an essential role for kinesin-13 during the blood stage. This previous study found that 5 kinesins (kinesin-4, kinesin-8B, kinesin-8X, kinesin-20, and kinesin-X4) are not essential for blood stage growth but provided no data for kinesin-5, kinesin-15, and kinesin-X3 [23].

Phenotypic analyses of the *kinesin*-knockout parasites, in comparison with the parental parasite (WTGFP), were carried out at various stages of the life cycle: in asexual blood stages, during male gametogenesis and the formation of exflagellation centres, during ookinete formation, in the number and size of oocysts, for the formation of sporozoites in oocysts and their migration to salivary glands, and for parasite transmission to the vertebrate host (**Fig 1D**). Taken together with previously published studies on kinesin-5, kinesin-8B, and kinesin-8X, only 2 knockout parasite lines (*Δkinesin-8B* and *Δkinesin-15*) showed a defect in the formation of male gametes (**Figs 1D and S5A**). *Δkinesin-8B* parasites produced no male gametes, as shown previously [20,21], while there was a significant decrease in male gamete formation in *Δkinesin-15* parasites (**Figs 1D and S5A**). Next, we analysed the zygote to ookinete transition (round to banana-shaped cells) after 24 h following gametocyte activation. Three parasite lines (*Δkinesin-8B*, *Δkinesin-15*, and *Δkinesin-20*) produced no or reduced numbers of ookinetes (**S5B Fig**). *Δkinesin-8B* parasites produced no ookinetes, as expected because there were no male gametes to fertilise the female gametes (**Figs 1D and S5B**) [20]. *Δkinesin-15* parasites produced significantly fewer male gametes, which would be expected to result in fewer ookinetes compared to WTGFP parasites (**Figs 1D and S5B**). In contrast, *Δkinesin-20* parasites exflagellated normally, and, therefore, loss of this kinesin must have a direct effect on ookinete formation (**S5B Fig**).

To assess the effect of kinesin gene deletions on oocyst development and infective sporozoite formation, 40 to 50 *Anopheles stephensi* mosquitoes were fed on mice infected with individual *kinesin*-knockout lines, and parasite development was examined. First, GFP-positive oocysts on the mosquito gut wall were counted at 7, 14, and 21 days post-infection (dpi). Three out of 8 kinesin-knockout lines showed defects in oocyst production; *Δkinesin-8B* parasites produced no oocysts as shown previously [20], while there was a significant reduction in *Δkinesin-15* and *Δkinesin-20* oocysts compared to WTGFP oocysts at 7 dpi and a further reduction by 14 and 21 dpi (**S5C Fig**). The adverse effects on ookinete production rather than a direct effect on oocyst development could explain this observation. Although there was no significant difference in the number of oocysts of other kinesin gene knockouts compared to WTGFP at 7 dpi, a significant reduction was observed for the *Δkinesin-8X* line at 14 dpi, which became more evident by 21 dpi (**S5C Fig**). Oocyst size was not affected in most of the lines that produced them; the only exception was *Δkinesin-8X* oocysts, which were approximately half the size of WTGFP oocysts at 14 dpi, and even smaller by 21 dpi (**S5D Fig**). Four out of 8 kinesin-knockout lines produced no or defective sporozoites; *Δkinesin-8B* and *Δkinesin-8X* produced no sporozoites, as reported earlier [18,20], while *Δkinesin-15* and *Δkinesin-20* lines had significantly reduced sporozoite numbers compared to control parental parasites (**Figs 1D and S5E**). These defects were mirrored in the salivary glands: For the *Δkinesin-8B* and *Δkinesin-8X* lines, no sporozoites were detected, as reported earlier [18,20], while *Δkinesin-15* and *Δkinesin-20* lines had a significantly reduced number. The *Δkinesin-5* parasite produced significantly fewer infective salivary gland sporozoites (**S5F Fig**) as reported previously [19]. However, although several kinesin gene-knockout lines exhibited defects in sporozoite production and reduced salivary gland infection, these sporozoites were still infectious to the mammalian host as observed with successful infection of new hosts in mosquito bite back experiments

(**Figs 1D** and **S5G**). In summary, for most of the kinesin gene-knockout *P. berghei* lines, there were clear developmental defects at specific stages of the life cycle within the mosquito vector.

## Apicomplexa-enriched kinesins have discrete locations during pellicle formation (-X3) and axoneme assembly (-X4) during sexual development

Previous bioinformatic analysis identified 2 divergent *Plasmodium* kinesins (kinesin-X3 and kinesin-X4) [5,18]; one of them (kinesin-X3) is restricted to the phylum Apicomplexa [18]. Kinesin-X4 is also restricted to Apicomplexa except that it is also present in the starlet sea anemone *Nematostella vectensis* [5]. The parasitic Apicomplexa are characterised by a specialised apical structural complex that coordinates the interaction with and penetration of host cells, and have a surface pellicle comprised of the plasma membrane and an underlying layer of fused flattened membrane vesicles of the inner membrane complex (IMC) with associated MTs [24,25]. To examine whether the kinesins are associated with these apicomplexan features, localisation by live cell imaging was performed. Kinesin-X3 and kinesin-X4 showed stage-specific expression during sexual development with a distinct location (**S3 Fig**). During zygote to ookinete differentiation, kinesin-X3 expression was restricted to one side of the cell in the early stages of development (stages I to III), suggesting an involvement in pellicle formation (**Fig 2A**). In later stages (stages IV to VI), the kinesin-X3 location became more distinct around the periphery of the ookinete. Monoclonal antibody (mAb) 13.1 conjugated with cy3 (red), which recognises the P28 protein on the surface of zygote and ookinete stages, stained these stages, and colocalised with kinesin-X3 (green) (**Fig 2A**), although kinesin-X3 was not present at the apical and basal ends of the developing ookinete (**Fig 2A**). The data suggest that kinesin-X3 is restricted to pellicle formation during ookinete and sporozoite stages in the mosquito.

Using real-time live cell imaging of male gametogenesis, the expression and location of kinesin-X4 (green) was compared with that of axonemal protein kinesin-8B (red) located on basal bodies and axonemes [20]. Kinesin-X4 showed a diffuse cytosolic distribution during early stages of male gametogenesis (1 to 3 min post-activation [mpa]) but no strong signal on the basal body tetrads labelled with kinesin-8B (red) (**Fig 2B**). However, at 4 to 6 mpa, the kinesin-X4 signal distribution changed to resemble linear structures, which were maintained in the later stages (8 to 10 mpa) and showed colocalisation with kinesin-8B (**Fig 2B**). These data suggest that kinesin-X4 is located on axonemes together with kinesin-8B during flagella formation in *Plasmodium* spp.

## Kinesin-8X and kinesin-5 are nuclear spindle kinesins associated with the kinetochore (NDC80) that bind centromeres

In our previous studies, we showed that 2 kinesins, kinesin-5 and kinesin-8X, are associated with spindles and restricted to the nucleus during most of the life cycle stages [18,19]. To further examine the spatiotemporal dynamics of these kinesins during spindle formation, chromosome segregation, and axoneme biogenesis during male gametogenesis, we crossed parasite lines expressing kinesin-8XGFP and kinesin-5GFP with lines expressing NDC80-Cherry, a kinetochore protein in the nucleus, and kinesin-8B-Cherry, an axonemal protein in the cytoplasm, and compared protein location by live cell imaging (**S6A Fig**). Both kinesin-8X and kinesin-5 (green) were colocalised with NDC80 (red) suggesting a role in mitotic spindle function and chromosome segregation (**Figs 2C** and **S6B**). On the other hand, neither kinesin-5 nor kinesin-8X showed any overlap with cytosolic kinesin-8B (red) during male gametogenesis (**S6C and S6D Fig**) confirming their restricted location within the nuclear compartment.

Kinetochores are multiprotein complexes assembled at the centromere of each chromosome, which mediate chromosome attachment to spindle MTs. Because kinesin-8X and

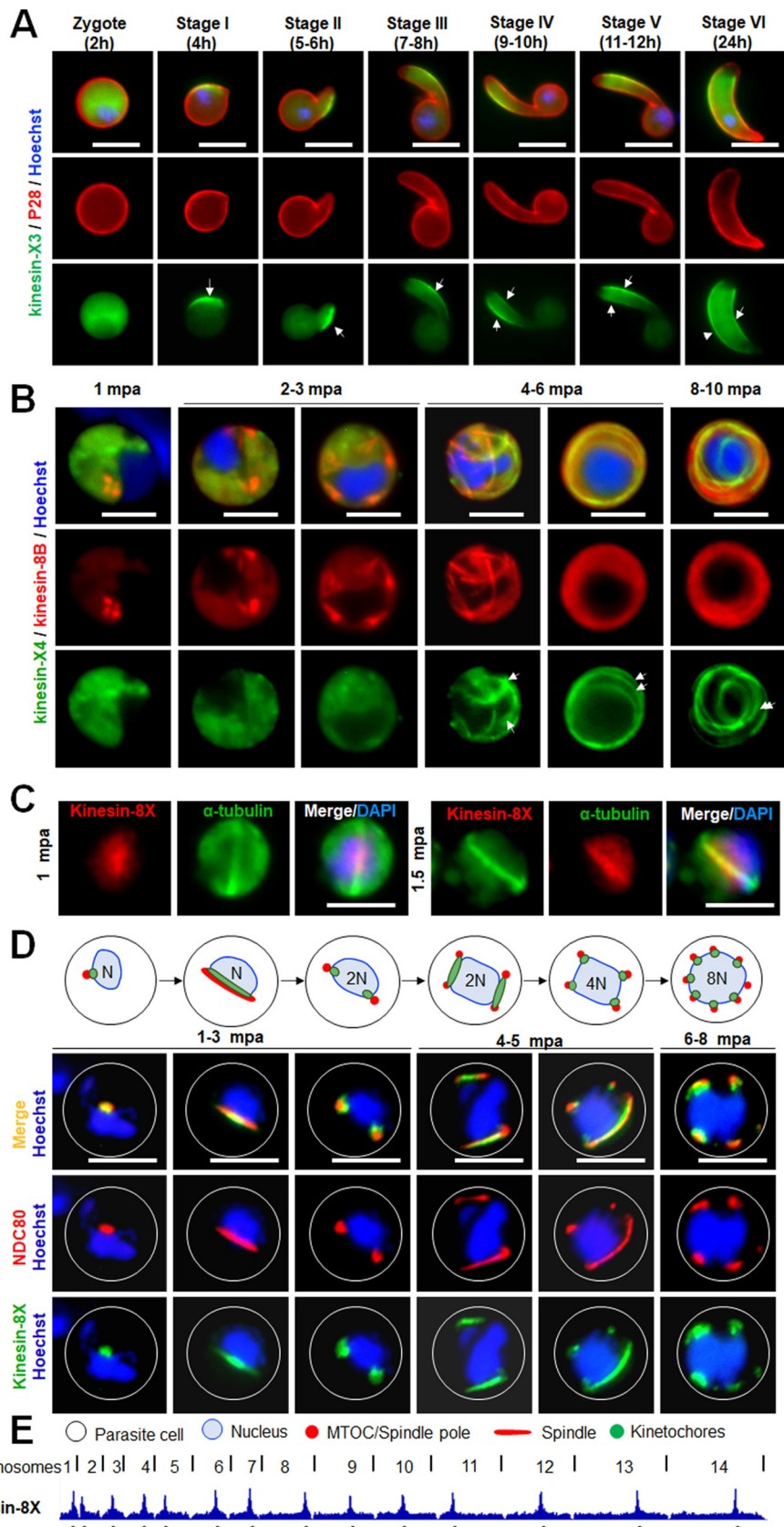

**Fig 2. Apicomplexa-enriched kinesins (kinesin-X3 and kinesin-X4) are located at the pellicle during ookinete differentiation and at axonemes in male gametogenesis, respectively, while nuclear kinesin (kinesin-8X) associates with the kinetochore at the centromere during male gamete formation. (A)** Live cell imaging showing temporal location of kinesin-X3 (green) associated with pellicle formation (arrows) during zygote to ookinete transition (2–24 h after fertilisation). A cy3-conjugated antibody, 13.1, which recognises the protein P28 on the surface of zygote and ookinete stages, was used to track these stages (red). DNA is stained with Hoechst dye (blue). Scale bar = 5 μm. **(B)** Live cell imaging shows the association of kinesin-X4 (green) with axoneme marker kinesin-8B (red) during male gametogenesis. Note in later stages, axonemes (arrows) are labelled with both markers. DNA is stained with Hoechst dye (blue). Scale bar = 5 μm. **(C)** Live cell imaging showing the dynamics of kinesin-8XGFP (green) along with kinetochore marker NDC80Cherry (red) during male gametogenesis. DNA is stained with Hoechst dye (blue); scale bar = 5 μm. A diagrammatic representation of spindle/spindle pole and kinetochore is shown in upper panels. **(D)** Indirect immunofluorescence assays showing colocalisation of Pbkinesin-8X (red) and α-tubulin (green) in activated male gametocytes. Scale bar = 5 μm. **(E)** ChIP-seq analysis of kinesin-8X and NDC80 during gametocyte stage. Lines on top are division points between chromosomes and circles on the bottom indicate locations of centromeres. mpa, min post-activation.

kinesin-5 showed colocalisation with kinetochore protein NDC80, we analysed further the binding of these proteins at the centromere DNA. We performed ChIP-seq experiments with parasites undergoing gametogenesis (6 mpa), using kinesin-8XGFP and kinesin-5GFP tagged parasites and GFP-specific antibodies. Strong ChIP-seq peaks for each chromosome were observed with these kinesins, indicating their binding sites. Binding was restricted to a region close to the previously annotated centromeres of all 14 chromosomes [26] and identical to those identified in *Plasmodium* Condensin and NDC80 studies [8,27] **(Figs 2D and S7)**. Together, live cell imaging and ChIP-seq analysis support the notion that kinesin-8X and kinesin-5 associate with kinetochores assembled at centromeres.

## Kinesin-20-GFP location reveals a ring-like structure during ookinete differentiation, and deletion of the kinesin-20 gene affects ookinete morphology and motility

In the initial phenotypic screen described above, the *Δkinesin-20* parasite did not produce normal ookinetes (**S5B Fig**) but did produce a few oocysts (**S5C Fig**), so we undertook a more in-depth analysis to investigate this further. First, we analysed the spatiotemporal profile of kinesin-20GFP expression during zygote to ookinete differentiation, using P28 as a cell surface marker. Live cell imaging showed a diffuse distribution of kinesin-20GFP in the female gametocyte and zygote (**Figs 3A and S2**). Subsequently, the intensity of kinesin-20GFP increased in the developing apical protuberance from the main cell body, and in later stages, especially at stages II and III that are about 6 to 8 h after gametocyte activation (**Fig 3A**). The protein appeared as a ring restricted to the junction of the main cell body and the protrusion that is characteristic of developing ookinetes during stage II to stage V, and then dispersed in mature ookinetes (stage VI), with a largely diffuse distribution like that in the zygote stage (**Fig 3A**).

Next, we examined the *Δkinesin-20* parasites to ascertain whether the zygote was formed and at what stage parasite development was blocked. The *Δkinesin-20* parasite developed a short protuberance in stage II like that of the WTGFP control (**Fig 3B**), but this protrusion developed into a bulbous structure rather than the characteristic banana-shaped ookinete (**Fig 3B**) and remained like this 24 h later when the banana-shaped WTGFP ookinete had fully differentiated (**Fig 3B and 3C**). Since the mature wild-type ookinete is a motile and invasive stage, we investigated the motility of the *Δkinesin-20* bulbous ookinete using a Matrigel substrate, as described previously [18,28]. There was a significant decrease in the motility of *Δkinesin-20* ookinetes compared with the WTGFP ookinetes that showed normal gliding motility (**Fig 3D and 3E and S1 and S2 Movies**).

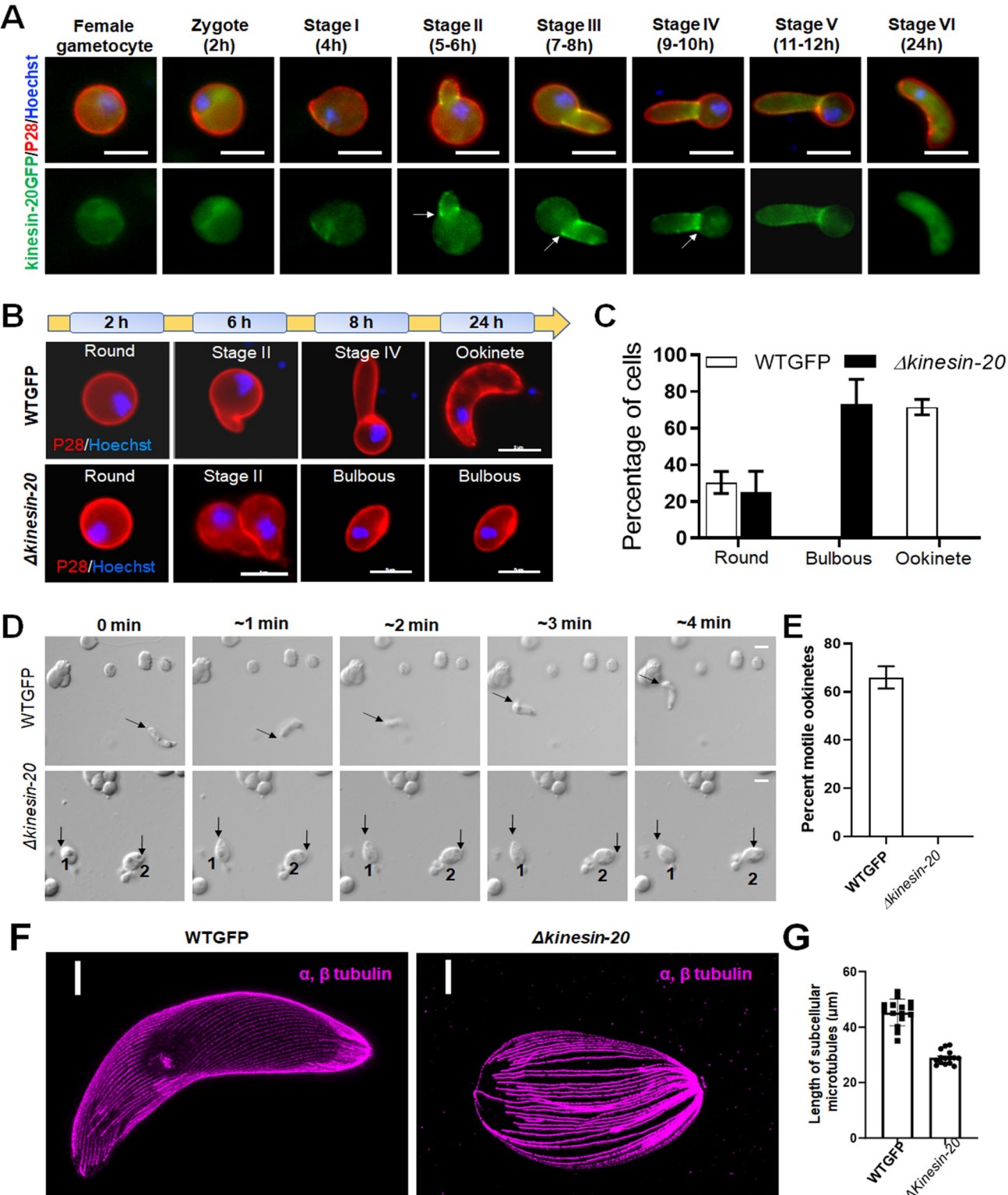

**Fig 3. Kinesin-20 makes a ring-like structure at the base of developing ookinete and is important for zygote to ookinete transformation and subsequent motility. (A)** Live cell imaging showing localisation of kinesin-20GFP (green) in the female gametocyte, zygote, and during ookinete transformation. Kinesin-20GFP accumulates at the neck of developing ookinete making a ring-like structure (indicated by arrows). Labelling with P28 was used to identify the surface of activated female gametes, zygotes, and ookinetes and to mark these stages (red). DNA is labelled with Hoechst dye (blue) **(B)**. Ookinete developmental stages for *Δkinesin-20* and WTGFP parasites. Ookinetes were identified using P28 and defined as those cells

differentiated successfully into elongated "banana-shaped" forms. Round cells are arrested zygotes or female gametocytes, while bulbous cells did not develop to banana-shaped ookinetes for *Δkinesin-20* parasites. DNA is stained with Hoechst dye (blue). Scale bar = 5 μm. **(C)** Ookinete conversion as a percentage of cells for *Δkinesin-20* and WTGFP parasites after 24 h. Mean ± SEM. *n* = 4 independent experiments. **(D)** Representative frames from time-lapse videos of motile WTGFP ookinetes and sessile ookinetes of *Δkinesin-20* in Matrigel. Black arrow indicates apical end of ookinete. Scale bar = 5 μm. **(E)** Ookinete motility for WTGFP and *Δkinesin-20* lines in Matrigel. More than 20 ookinetes were examined for each line. Mean ± SEM. *n* = 3 independent experiments. **(F)** Representative confocal images of expanded ookinetes of WTGFP and bulbous cells of *Δkinesin-20* lines stained for α- and β-tubulin (magenta) showing subpellicle MTs. Scale bar = 1 μm. **(G)** The MT lengths were measured from the images of ookinetes stained to reveal α and β tubulins obtained by expansion microscopy. The bar diagram shows the length of subpellicular MTs in bulbous *Δkinesin-20* ookinetes compared to WTGFP ookinetes. Mean ± SEM. *n* = 3 independent experiments. Underlying data are provided in the Supporting information as S1 Data. MT, microtubule.

Although *Δkinesin-20* parasites made no morphologically normal and motile ookinetes, nevertheless, they formed a few oocysts that produced sporozoites (**S5 Fig**). This would point to the mutation specifically affecting motility but not invasion of the mosquito gut. It is possible that a few immotile ookinetes may contact the mosquito gut wall during gut turbulence and these are able to invade and initiate oocyst formation. The *Δkinesin-20* sporozoites were morphologically like WTGFP parasites, so we examined their motility on Matrigel as described previously [28,29]. The motility of *Δkinesin-20* sporozoites was like that of WTGFP sporozoites (**S8A and S8B Fig** and **S3** and **S4 Movies**), suggesting that the defect in *Δkinesin-20* parasite shape and motility is limited to ookinete development.

To examine whether the expression of other kinesin genes is misregulated in *Δkinesin-20* gametocytes at 2 h post activation, we performed quantitative real-time PCR (qRT-PCR) analysis of them. We wished to check expression of other kinesins located in the cytoplasmic compartment of ookinete stages, e.g., kinesin-X3 and kinesin-13 expecting that expression of other kinesins coordinated with kinesin-20 would be modulated. However, there was no significant change in expression of any other kinesin gene in *Δkinesin-20* parasites compared with WTGFP parasites (**S8C Fig**).

To determine genome-wide changes in transcription, we performed RNA-seq analysis of the *Δkinesin-20* gametocytes at 2 h post-activation, because kinesin-20 expression starts in female gametocytes and continues into later stages. *Plasmodium* displays translational repression in female gametocytes with derepression only after fertilisation and translation of many proteins in the zygote [30]. We wished to identify any role of kinesin-20 in translational derepression. The *kinesin-20* deletion in the *Δkinesin-20* strain was confirmed, since no significant reads mapped to this gene locus (**S8D Fig**). We did not find any significant change in transcriptome due to *kinesin-20* deletion except 1 gene was up-regulated, and 16 genes were down-regulated (**S8E Fig** and **S1A Table**). Most of the differentially expressed genes belong to *pir* and *fam* gene clusters located in telomeric and subtelomeric regions of chromosomes. Two of the genes (PBANKA_1465700 and PBANKA_0200600) most down-regulated in *Δkinesin-20* gametocytes belong to the *fam* gene cluster and were described as female gametocyte-specific genes in a recent study [31]. This suggests a link between kinesin-20 function and female gametocyte development but no role of kinesin-20 in translational derepression.

## Ultrastructure analysis of *Δkinesin-20* ookinetes reveals disorganised subpellicular microtubules

The unusual size and shape of *Δkinesin-20* ookinete led us to perform high-resolution and ultrastructure analysis of the bulbous ookinetes using ultrastructure expansion and electron microscopy. By expansion microscopy using tubulin-specific antibody, we observed a marked reduction in MT length (**Fig 3F**): The length of MTs in *Δkinesin-20* bulbous ookinetes was decreased by approximately 40% compared to those of WTGFP parasites, reflecting a

reduction in the overall length of the ookinete (**Fig 3G**). This difference was confirmed by electron microscopy observations (**Fig 4**).

In an ultrastructural comparison of WTGFP and *Δkinesin-20* ookinetes, the most obvious difference was the shape of the cell. In longitudinal section, the WTGFP ookinetes were elongated with a crescentic outline (**Fig 4Aa**), and, in contrast, *Δkinesin-20* ookinetes were less elongated and had a more bulbous appearance (**Fig 4Ab and 4Ac**). In WTGFP parasites, the distribution of subcellular organelles was ordered with most micronemes in the apical cytoplasm, a more centrally located crystalline body and a posterior nucleus (**Fig 4Aa**). In contrast, early *Δkinesin-20* ookinetes had a large central nucleus with a few dense granules but lacked both micronemes and a crystalline body (**Fig 4Ab**). Others that appeared more mature, possessed similar organelles (micronemes, crystalline body, nucleus) to those of the WTGFP but differed from the control in having more randomly distributed micronemes (**Fig 4Ac**).

Due to the differences in cell shape, the apical complex and pellicle were examined in detail. When the apical complex was examined in longitudinal (**Fig 4Ad and 4Ae**) and cross (**Fig 4Af and 4Ag**) section, the complex nature of the structure was revealed. Interestingly, WTGFP and *Δkinesin-20* ookinetes displayed an identical substructure (**Fig 4Ad**, **4Ae**, **4Af and 4Ag**). In longitudinal sections of the central apex region, 3 conoidal rings could be identified underneath the plasma membrane. A unique substructure of the ookinete is the apical collar, which represents a cone-like structure embedded between the MTs and IMC of the pellicle [13]. The outer region of the collar is electron dense and appears to be fused to the IMC, which is interrupted at the apical end to form apical polar ring 1 (**Fig 4Ad–4Ag**). The inner aspect of the collar is more electron lucent and in close contact with the subpellicular MTs (**Fig 4Ad–4Ag**). The apical ends of the MTs are attached to a ring forming apical polar ring 2 (**Fig 4Ad and 4Ae**). For a detailed review of the apical complex, see Koreny and colleagues [13]. Approximately 50 subpellicular MTs emanate from polar ring 2 and run longitudinally beneath the collar (**Fig 4Af and 4Ag**) and then beneath the IMC of the pellicle (**Fig 4Ah and 4Ai**). In the region of the collar, MTs were evenly distributed in both WTGFP and *Δkinesin-20* parasites (**Fig 4Af and 4Ag**). However, in more posterior sections, while there continued to be an even distribution of MTs in close contact with the IMC in the WTGFP ookinete (**Fig 4Ah**), in the *Δkinesin-20* parasite, there were areas where there was uneven distribution, clumping, and detachment of MTs from the IMC (**Fig 4Ai**). A schematic of the ookinete apical end is given in **Fig 4B**. It shows that substructures of the apical complex (i) are identical in both WTGFP and *Δkinesin-20* parasite. The MTs in the region of the collar (ii) are evenly distributed, but they are unevenly distributed, clumped, and detached in posterior sections (iii) of *Δkinesin-20* ookinetes.

### Kinesin-13 associates with kinetochore marker NDC80 at all proliferative stages of the life cycle and its knockdown affects male gamete formation

Since the kinesin-13 gene was the only kinesin gene essential for blood stage schizogony and could not be disrupted in our genome-wide screen (**Figs 1D and S5**), we made a detailed analysis of the protein. We observed both a diffuse cytoplasmic distribution and a distinct nuclear pattern of kinesin-13GFP in all proliferative stages as shown in **Figs 1C and S2**. We performed live cell coimaging of kinesin-13GFP and the NDC80-cherry kinetochore marker after crossing the 2 transgenic parasite lines to observe kinesin13-GFP dynamics during chromosome segregation in various developmental stages. There was colocalisation of kinesin-13 and NDC80 at all proliferative stages (**Fig 5**), for example, during the schizogony and sporogony endomitotic stages (**Fig 5A and 5B**). In the sexual cells, during the rapid mitosis of male gametogenesis, there was partial colocalisation of kinesin-13 and NDC80, but a substantial amount

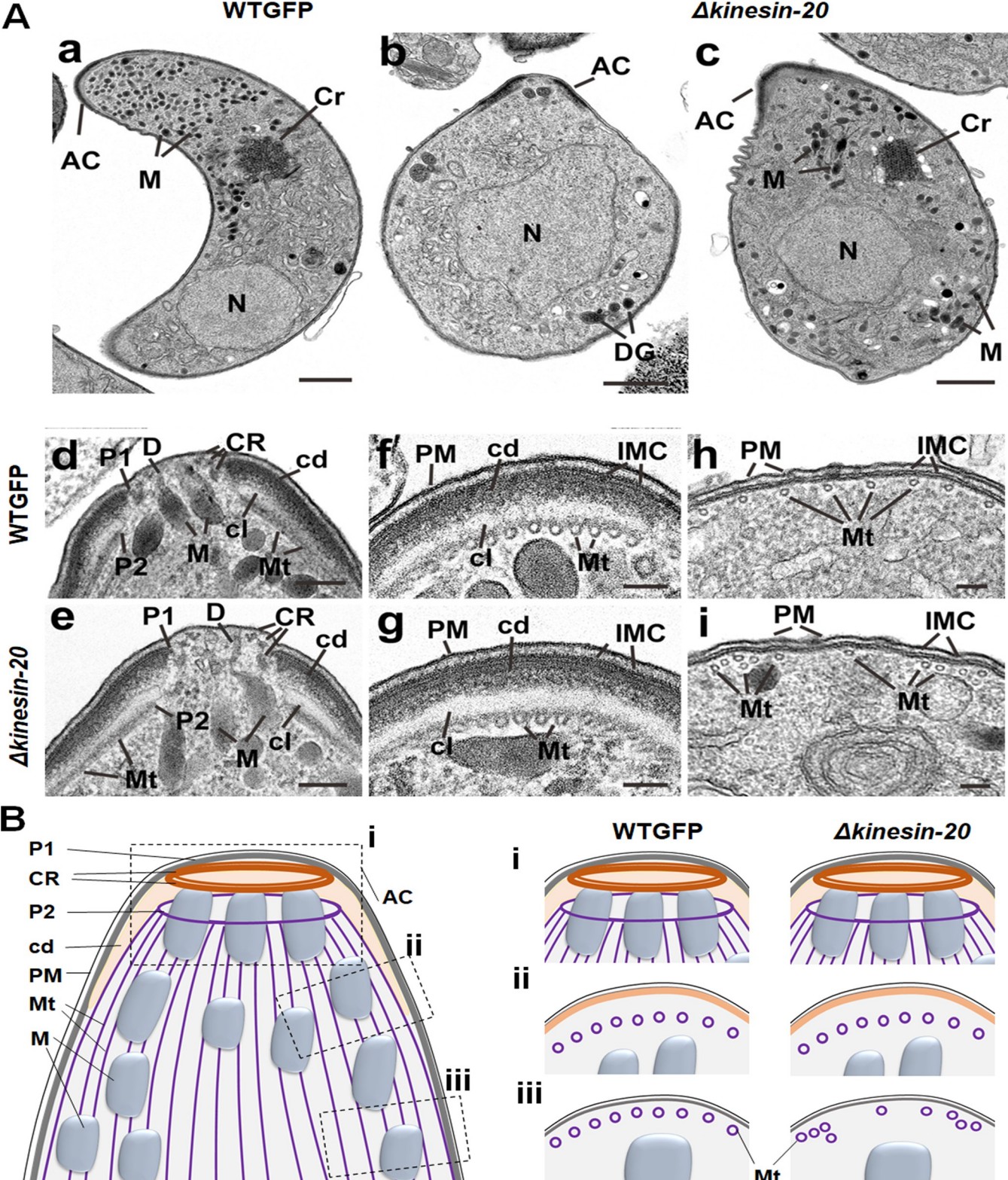

**Fig 4. Ultrastructural analysis of WTGFP and Δ*kinesin*-20 ookinetes.** (A) Electron micrographs of WTGFP (a, d, f, h) and Δ*kinesin*-20 (b, c, e, g, i) ookinetes. Bars represent 1 μm (a-c) and 100 nm (d-i). (a) Longitudinal section of a crescentic shaped WTGFP ookinete showing the apical complex (AC) with numerous micronemes (M) in the anterior cytoplasm and a more posteriorly located crystalline body (Cr) and nucleus (N). (b) Longitudinal section through an immature Δ*kinesin*-20 ookinete displaying a bulbous shape. Within the cytoplasm, the AC plus an irregular shaped N and a few irregularly arranged dense granules (DGs) can be identified. (c). Longitudinal section through a more mature Δ*kinesin*-20 ookinete showing a bulbous shaped cell. Within the cytoplasm, the AC, N, and

Cr can be identified but note the M appear to be distributed throughout the cytoplasm. (d, e) Details of longitudinal sections through the apical complex of WTGFP (d) and *Δkinesin-20* (e) ookinetes showing similar substructures consisting of 3 conoidal rings (CRs), an anterior polar ring 1 (P1) formed by the IMC and a second polar ring (P2) representing the initiation site for the subpellicular microtubules (Mt). The polar rings are separated by a collar consisting of an outer electron dense layer (cd) and an inner electron lucent layer (cl). Note the M with ducts (D) direct to the anterior plasmalemma (PM). (f, g) Cross section through the periphery of the anterior complex of a WTGFP (f) and a *Δkinesin-20* (g) parasite showing similar substructure consisting of the outer PM and the underlying IMC, which appears fused to the outer electron dense (cd) region of the apical collar while the more electron lucent inner region is in close contact with subpellicular Mt. (h, i) Cross section of the pellicle posterior to the apical complex consisting of the outer PM and the underlying IMC. Note that while the subpellicular Mt in the WTGFP parasite (h) are evenly spaced, those of the *Δkinesin-20* (i) show irregular spacing and some clumping. (B) A schematic diagram of ookinete showing the defects in Mt organisation posterior to apical complex. cd, dense layer; cl, lucent layer; Cr, crystalline body; CR, conoidal ring; D, ducts; DG, dense granule; IMC, inner membrane complex; M, micronemes; Mt, microtubules; N, nucleus; PM, plasmalemma; P1, polar ring 1; P2, polar ring 2.

of kinesin-13 was also located in the cytosolic compartment (**Fig 5C**). In the meiotic stage during ookinete development there was clear colocalisation of kinesin-13 and NDC80 (**Fig 5D**). At the start of meiosis (2 h after zygote formation), there was one strong nuclear focus, and at the end of ookinete formation, there were 3 to 4 colocalised foci (**Fig 5D**). To further examine the location of kinesin-13, we used ultrastructure expansion microscopy to examine gametocytes activated for 15 min and then compared its location with that of tubulins. Kinesin-13 (green) was observed to colocalise with α/β tubulin (magenta) suggesting a location on axonemes and spindles (**Fig 6A**). Further analysis of the male gamete by fixed immunofluorescence microscopy revealed a colocalisation of kinesin-13 (green) with tubulin (red) (**Fig 6B**).

Since kinesin-13 is essential for the asexual blood stage, and the gene could not be deleted, we applied 2 conditional genetic knockdown strategies to evaluate its role at other proliferative stages within the mosquito vector. First, we used an auxin-inducible degron (AID) system to try and study the effect of rapid kinesin-13 degradation in gametocytes. We tagged the endogenous kinesin-13 gene with an AID/HA epitope tag (**S9A Fig**) to degrade the fusion protein in presence of auxin [32] and successfully generated kinesin-13-AID parasite lines as shown by integration PCR (**S9B Fig**) but could not deplete kinesin-13 protein by auxin treatment (**S9C Fig**). Next, we used a promotor trap double homologous recombination (PTD) approach, inserting the *clag* promotor at the 5′ end of kinesin-13, and generated the conditional knockdown parasite: P*clag*kinesin-13 (kinesin-13PTD) (**S9D Fig**). *clag* (cytoadherence-linked asexual gene) is highly expressed in asexual blood stages, but largely silent during stages of sexual differentiation, including gametocytes and ookinetes [33]. The successful insertion was confirmed by diagnostic PCR (**S9E Fig**) and qRT PCR showed a significant down-regulation of kinesin-13 gene transcripts in *kinesin-13PTD* gametocytes, when compared to WTGFP gametocytes (**Fig 6C**).

The phenotype of the *kinesin-13PTD* modification was examined during various stages of the parasite life cycle. Parasite proliferation in asexual blood stages was not affected, but during male gametogenesis, exflagellation was markedly reduced and very few male gametes were produced by *kinesin-13PTD* parasites compared to WTGFP parasites (**Fig 6D**). Zygote development and ookinete differentiation were severely affected, and no clear banana-shaped ookinetes were observed (**Fig 6E**). Subsequent stages in the mosquito were also affected significantly and no oocyst or sporozoite formation was observed (**S9F Fig**), and consequently no transmission of *kinesin-13PTD* parasites occurred, as shown by mosquito bite back experiments (**S9G Fig**).

## Global transcriptomic analysis of *kinesin-13PTD* parasites shows misregulation of transcripts for gene clusters involved in axoneme assembly and chromosome dynamics

To examine the transcript level of other kinesins in *kinesin-13PTD* gametocytes, we performed qPCR for all the 9 kinesins and found that some, like kinesin-8B, kinesin-8X, kinesin-15, and kinesin-20, were down-regulated (**Fig 6F**).

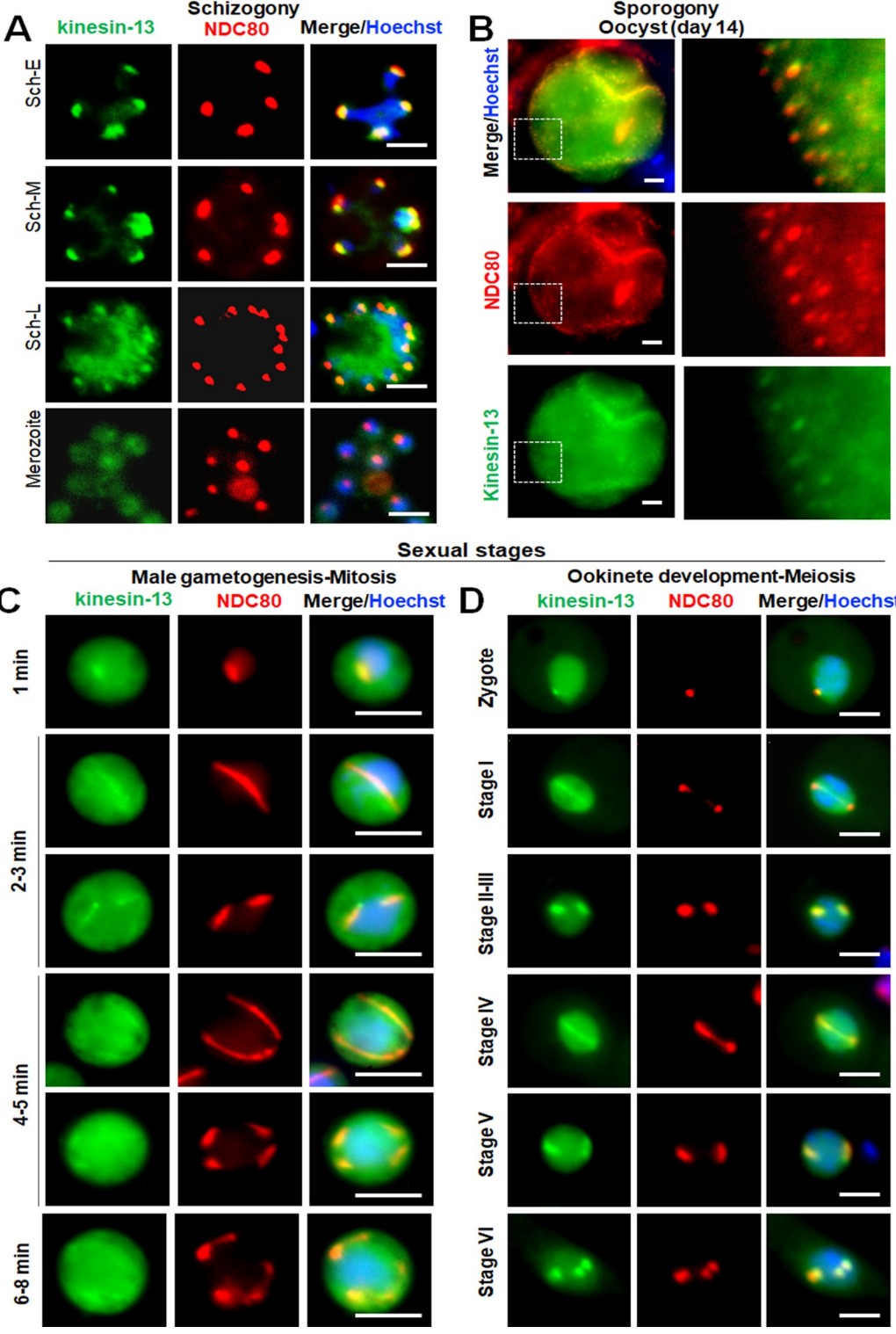

**Fig 5. Kinesin-13 associates with kinetochore marker, NDC80, during various proliferative stages of parasite life cycle.** Live cell imaging showing location of kinesin-13GFP (green) with respect to kinetochore marker NDC80Cherry (red) during asexual (**A**, **B**) and sexual (**C**, **D**) proliferative stages. Asexual proliferative stages include endomitosis during blood schizogony and sporogony. Sexual stages include endomitosis during male gametogenesis and meiosis during ookinete development. Kinesin-13GFP (green) shows a diffuse distribution in the cytoplasm, together with strong foci at

different stages of development that colocalise with NDC80Cherry (red). DNA is stained with Hoechst dye (blue). Scale bar = 5 μm.

Since transcripts of other kinesins were affected, global transcription was investigated by RNA-seq analysis of *kinesin-13PTD* gametocytes at 0 and 15 mpa, representing times point before the start of male gametogenesis and just after male gamete formation (exflagellation), respectively. *Kinesin-13* down-regulation in *kinesin-13PTD* gametocytes, relative to WTGFP gametocytes, was confirmed by RNA-seq analysis, showing the lack of reads for this locus (**Fig 6G**). In addition to reduced kinesin-13 expression, 34 other genes were significantly down-regulated and the expression of 152 genes was significantly up-regulated in *kinesin-13PTD* gametocytes before activation (at 0 mpa) (**Fig 6H** and **S1B Table**). Similarly, the expression of 22 genes was significantly down-regulated and the expression of 329 genes was significantly up-regulated in *kinesin-13PTD* gametocytes after 15 min activation (**Fig 6I** and **S1C Table**). Bioinformatic analysis of these differentially regulated genes revealed 2 important clusters of genes that were affected, including those coding for proteins involved in axoneme assembly, glideosome assembly, and chromosome dynamics (**Fig 6J**).

### High-resolution ultrastructure analysis of *kinesin-13PTD* parasites identifies defects in spindle assembly and axoneme MT of gametocytes and the subpellicular MT of ookinetes

Phenotypic analysis of *kinesin-13PTD* parasites revealed defects in male gamete and ookinete formation, and, therefore, we performed comparative high-resolution image analysis of *kinesin-13PTD* and WTGFP gametocytes and ookinetes. Ultrastructure expansion microscopy revealed that both spindle and axoneme MTs were disorganised, and no clear flagella were visible at 4 to 5 or 15 min after male gametocyte activation, respectively (**Fig 7A**). Disorganised MT were also observed in the defective *kinesin-13PTD* zygotes/ookinetes in comparison to the corresponding WTGFP parasites (**Fig 7A**).

These data were substantiated by electron microscopy analysis of male gametocytes activated for 6 or 15 min (**Fig 7B**). The electron micrographs showed that in both WTGFP and *kinesin-13PTD* parasites at 6 mpa, most male gametocytes were at an early stage of development, with a spherical appearance and a large central nucleus (**Fig 7Ba and 7Bc**). In WTGFP gametocytes, several nuclear poles (also known as spindle poles) were observed while the cytoplasm contained several normal 9+2 axonemes and some abnormal axonemes (**Fig 7Ba and 7Bb**). Although in the *kinesin-13PTD* gametocytes few nuclear poles were observed (**Fig 7Bc**), the major difference was in the cytoplasm, where there were collections of free single and double MTs plus several partially organised into axoneme-like structures (**Fig 7Bd**) while 9+2 axonemes were very rare. At 15 mpa, in the WTGFP samples, several late stages were observed with evidence of exflagellation and protruding microgametes (**Fig 7Be**) and several free male gametes complete with nucleus and flagellum were observed (**Fig 7Bf and 7Bg**). In contrast, in the *kinesin-13PTD* parasites, most male gametocytes were still at an early stage, and in the few at a later stage, the nucleus displayed clumps of chromatin (**Fig 7Bh**) with a few examples of protrusions of MTs from the plasma membrane (**Fig 7Bi**), but with no evidence of flagella formation or free male gametes.

### Discussion

*Plasmodium* spp. have a complex life cycle involving 2 hosts. They invade tissues and cells in diverse environments and have evolved a series of cellular shapes and sizes, with several

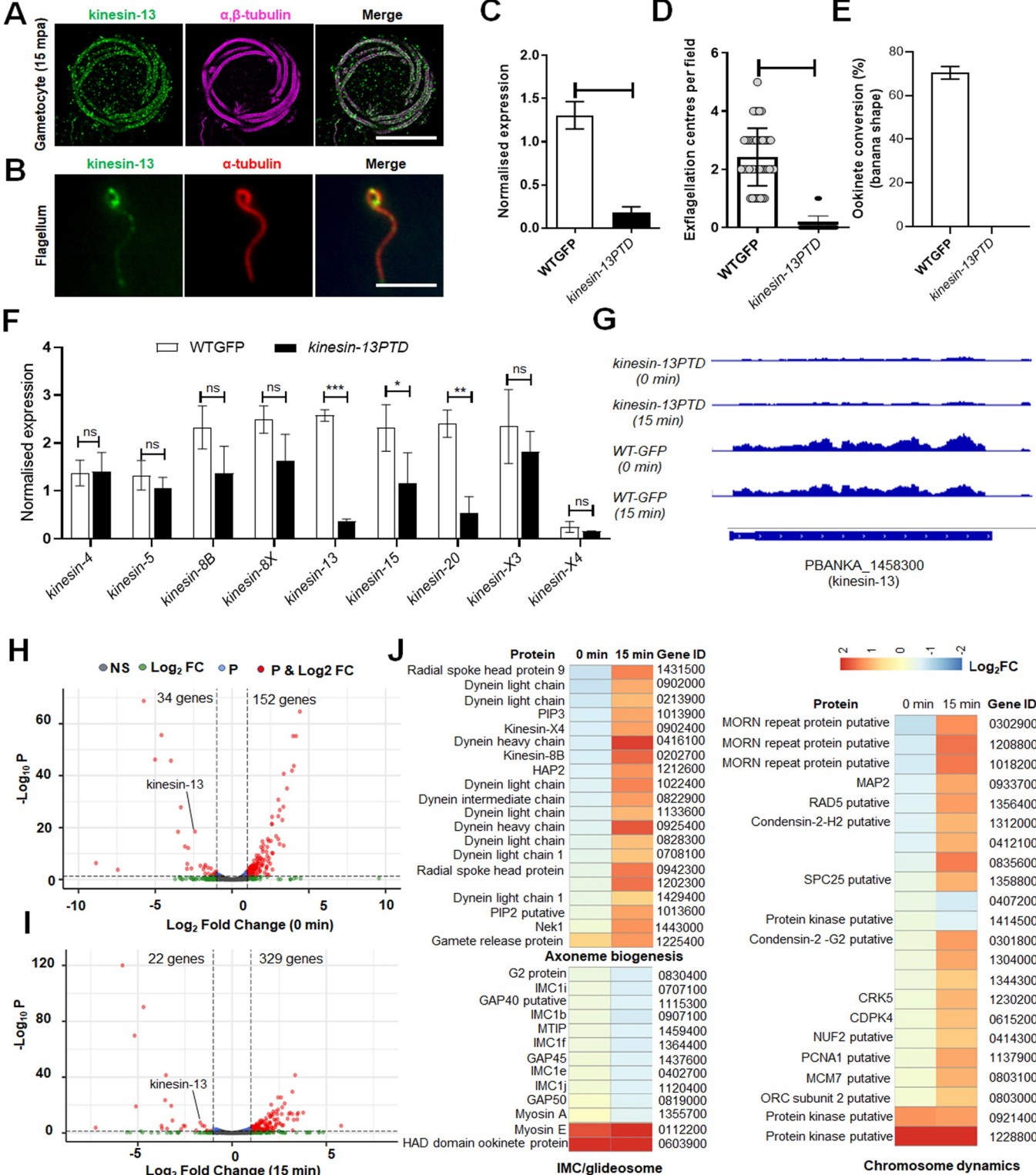

**Fig 6. Kinesin-13 associates with axonemes in male gametocytes and gametes and its knockdown affects male gamete formation. (A)** Expansion microscopy showing colocalisation of kinesin-13 (green) with α/β tubulin staining (purple) in gametocytes activated for 15 min. Scale bar = 5 μm. **(B)** Indirect immunofluorescence assay showing the colocalisation of kinesin-13 (green) and tubulin (red) staining in male gamete. Scale bar = 5 μm. (C) qRT-PCR analysis showing down-regulation of kinesin-13 gene in *kinesin-13PTD* parasites compared to WTGFP. Mean ± SEM, *n* = 3 independent experiments. (D) Exflagellation centres per field at 15 mpa. *n* = 3 independent experiments (>20 fields per experiment). Error bar, ±SEM. (E) Percentage ookinete conversion

from zygote. *n* = 3 independent experiments (>80 cells). Error bar, ±SEM. (F) qRT-PCR analysis of transcripts for other kinesin genes in *kinesin-13PTD* compared to WTGFP parasites. *n* = 3 independent experiments. Error bar, ±SEM. (G) RNA-seq analysis showing depletion of kinesin-13 transcript in *kinesin-13PTD* parasites. (H) Volcano plot showing differentially regulated genes in *kinesin-13PTD* gametocytes before activation (0 min). (I) Volcano plot showing differentially regulated genes in *kinesin-13PTD* gametocytes activated for 15 min. (J) Heat maps showing differential regulation of various genes involved in axoneme biogenesis, IMC/glideosome and chromosome dynamics. *$p \leq 0.05$, **$p \leq 0.01$, and ***$p \leq 0.001$. Underlying data are provided in the Supporting information as S2 Data. IMC, inner membrane complex; mpa, min post-activation; qRT-PCR, quantitative real-time PCR.

distinct morphological forms with cellular polarity and gliding motility for invasion, and cellular proliferation underpinned with an atypical mode of cell division [8–10]. Many of these processes are governed by MTs and MT-based motor proteins like kinesins [18]. In many organisms including *Plasmodium* spp., MTs form different structural frameworks such as the spindle assembly during cell division, the axonemes of cilia and flagella, and a cytoskeletal scaffold to establish and maintain cell polarity, cell shape, intracellular transport, and cell motility [34,35]. Kinesins regulate the organisation and function of MTs, using them as a track for movement or regulating their dynamics during cellular processes [1,6]. *Plasmodium* spp. are evolutionarily divergent unicellular eukaryotes with genomes that encode 9 kinesins including two that are Apicomplexa enriched (kinesin-X3 and kinesin-X4) and which lack genes for 3 classical kinesins (kinesin-1, kinesin-2, and kinesin-3) normally important in intracellular transport [5,18].

The expression and subcellular location of each *Plasmodium* kinesin provides important information about their potential role. Kinesin-13 is the most widely expressed of these motors in all proliferative and invasive stages, with diverse cytoplasmic locations including axonemes in male gametocytes and at the apical end of the differentiating ookinete in addition to its association with the nuclear spindle apparatus. A similar diverse set of kinesin-13 locations has been reported in other eukaryotes [36–38], highlighting its importance for various MT-associated biological processes in *Plasmodium* spp.

The cytoplasmic location of 3 male gametocyte-specific kinesins (kinesin-8B, kinesin-15, and kinesin-X4) and kinesin-13 suggests their role in rapid axoneme assembly and flagellum formation in *Plasmodium* spp. The axonemes are assembled directly in the male gametocyte cytoplasm—there is thus no requirement for transport of building materials by the intraflagellar transport (IFT) mechanisms common in many other eukaryotes [39–41]. Consistently, the IFT transport-associated kinesin-2 is absent from *Plasmodium*, further reinforcing the operation of noncanonical axoneme assembly mechanisms [42]. In good agreement with our findings, a previous *P. berghei* male gamete proteome study, 3 kinesins (kinesin-8B, kinesin-13, and kinesin-15) were identified and proposed to have an important role in axoneme assembly [43]. In a recent study on the regulation of *P. berghei* male gametogenesis, many of the identified phospho-regulated proteins had motor activity and included most of the *Plasmodium* kinesins [44]. The expression of 6 kinesins (kinesin-5, kinesin-8B, kinesin-8X, kinesin-13, kinesin-15, and kinesin-X4) in gametocytes and their location in either nucleus (kinesin-5 and kinesin-8X), cytoplasm (kinesin-8B, kinesin-15, and kinesin-X4), or both (kinesin-13) suggest the importance of these kinesins in male gametogenesis and thus parasite transmission.

Kinesin-X3 and kinesin-X4, which are largely Apicomplexa specific, show an interesting location on ookinete and sporozoite surface pellicle (-X3) and axonemes during male gametogenesis (-X4), respectively. These kinesins seem to have evolved in Apicomplexa for different MT-based structures such as the axoneme of flagella and a cytoskeletal scaffold to establish and maintain cell polarity, shape, and motility. The absence of kinesin-X3 from the apical end highlights that it is not a part of the conoidal complex [13,45]. Kinesin-20 has a ring-like location at the junction between the protrusion and the main cell body of the developing ookinete, suggesting a role in formation of the IMC/subpellicular MT complex and defining the size and

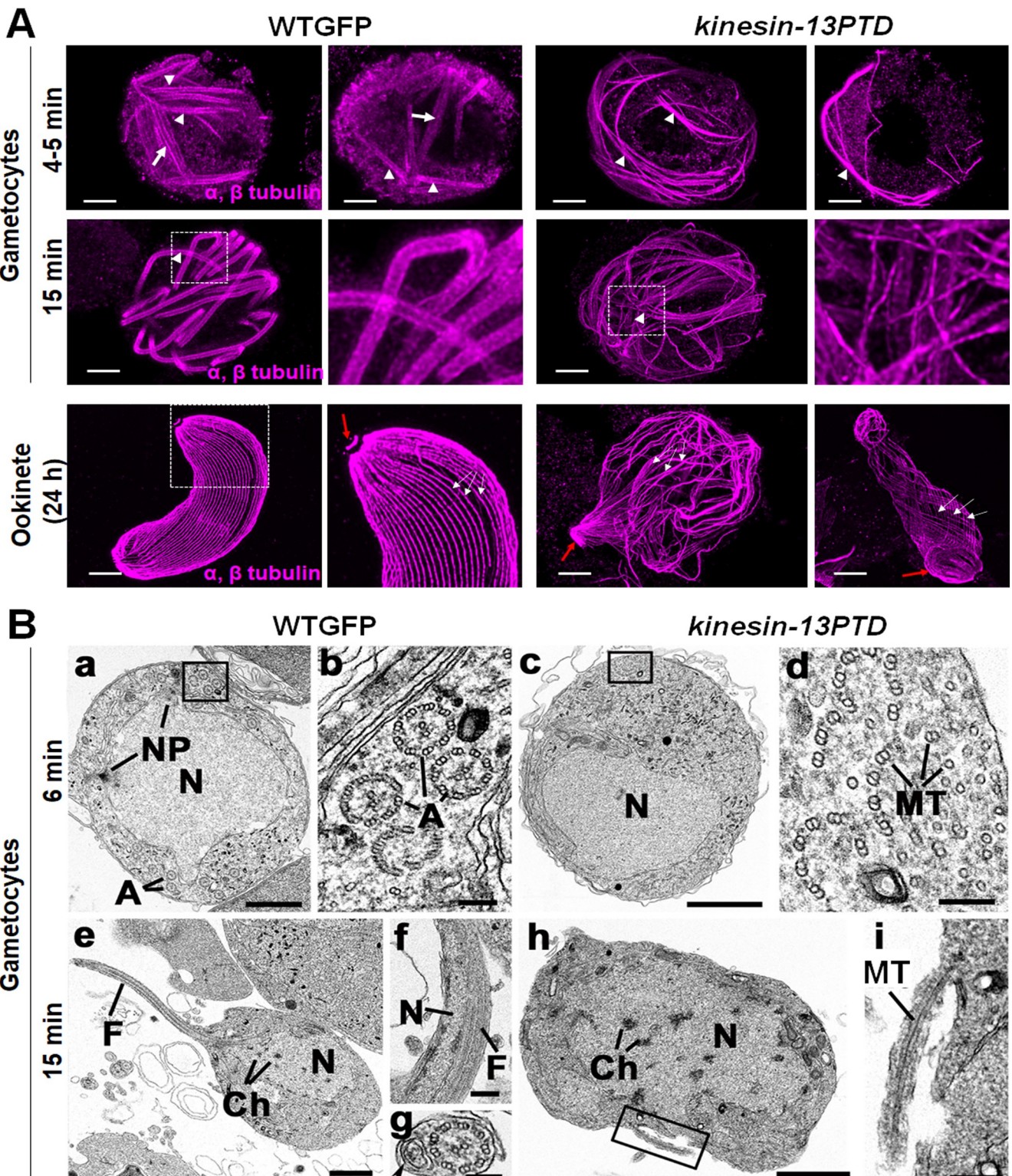

**Fig 7. Ultrastructural analysis of WTGFP and *kinesin-13PTD* gametocytes and ookinetes. (A)** Representative confocal images of expanded gametocytes of WTGFP and *kinesin-13PTD* lines stained for α- and β-tubulin (magenta) showing labelling of spindle (arrow) and axonemal MTs (arrowhead) at 4–5 min and axonemal MTs (arrowhead) at 15 mpa. Similarly, representative confocal images of expanded ookinetes of WTGFP and *kinesin-13PTD* lines stained for α- and β-tubulin (magenta) showing well-organised subpellicular MTs (white arrows) and apical tubulin ring (ATR, red arrows) in WTGFP ookinetes and disorganised MTs in *kinesin-13PTD* ookinetes. Scale bars = 1 μm. **(B)** Electron micrographs of *P. berghei* male gametogenesis of WTGFP **(a,**

**b, e, f, g)** and *kinesin-13PTD* (**c, d, h, i**) at 6 mpa (**a–d**) and 15 mpa (**e–i**). Bars represent 1 μm in **a, c, e, h** and 100 nm in other micrographs. **(a)** Early stage WTGFP showing the large central nucleus (N) displaying 2 nuclear poles (NPs) with cytoplasm containing several axonemes (A). **(b)** Detail of the cytoplasm illustrating several normal 9+2 axonemes (A) and abnormal axonemes. **(c)** Early-stage *kinesin-13PTD* gametocytes showing the central nucleus (N) but the cytoplasm lacked visible axonemes. **(d)** Detail of the enclosed area in **c** showing randomly arranged free single and duplet MTs with some forming partial axonemes. **(e)** Late stage WTGFP showing a flagellum (F) of a developing microgamete emerging from the male gametocyte (exflagellation). **(f)** Detail of a longitudinal section through a free male gamete showing the nucleus (N) and flagellum (F). **(g)** Detail of a cross section through a male gamete showing the 9+2 flagellum and the nucleus (arrowhead). **(h)** Late-stage *kinesin-13PTD* gametocyte showing the central nucleus (N) with clumps of chromatin (Ch) but an absence of any 9+2 axonemes. **(i)** Detail of the enclosed area in **h** showing a microtubule (Mt) enclosed by the plasmalemma emerging from the male gametocyte. Ch, chromatin; mpa, min post-activation; Mt, microtubule; N, nucleus; NP, nuclear pole.

shape of the cell and then disappears from the mature cell. *Plasmodium* Myosin F has a similar location in early stages of ookinete development [29] suggesting the existence of an actomyosin contractile ring that might be regulated by kinesin-20.

Genetic analysis revealed that most of the kinesins (8 out of 9) have their most important roles in transmission stages within the mosquito, where there are substantial changes in cell size, morphology, and invasive properties, which may be regulated by MTs and associated proteins. For example, the results of our ultrastructural analysis of *Δkinesin-20* parasites show that loss of this kinesin affects the development of ookinete shape and size. MT length, number, and association with the IMC are crucial to determine the size, shape, and motility of certain *Plasmodium* spp. stages [35]. We show that kinesin-20 regulates the length and arrangement of subpellicular MTs of developing ookinetes. Subpellicular MTs along with IMC proteins maintain ookinete polarity and morphology in *Plasmodium* spp. [25]. IMC1b-deficient ookinetes display abnormal cell shape and reduced gliding motility [46] similar to the properties of *Δkinesin-20*, and a similar phenotype was observed in a recent study showing that palmitoylation of IMC subcompartment proteins (ISPs) regulates the apical subpellicular MT network in ookinetes and affects their elongation [47]. ISPs also maintain the polar location of guanylate cyclase beta (GCβ)/CDC50A complex at the IMC, essential for ookinete gliding [48]. PPKL-deficient parasites also have a defective MT organisation and abnormal shaped ookinetes [49], but the *Δkinesin-20* phenotype is slightly different, with no defect in the apical ring, which serves as an MT organising centre for subpellicular MTs, and similar to what was found for phosphodiesterase-δ (*pdeδ*)-deficient ookinetes, which lack this enzyme involved in cyclic GMP signalling [50]. The kinesin-20GFP location suggests there is a ring-like structure at the junction of the protrusion and cell body, which defines the developing ookinete shape and diameter, while apical polarity guides ookinete size and differentiation. An actomyosin contractile ring is present in elongation of embryonic cells of *Ciona intestinalis*, a primitive chordate [51]. The assembly and organisation of an actomyosin contractile ring during cytokinesis is highly dynamic and contains, in addition to actin and myosin, other proteins that regulate actin nucleation, cross-linking, and myosin activity [52,53]. In *Plasmodium* spp. and other members of Apicomplexa, a similar contractile ring has been reported as required for cytokinesis [54,55] but an involvement in cell elongation is unknown. Kinesin-20 could be a protein that regulates contractile ring function in cell elongation during ookinete development.

The in-depth structural analysis of *kinesin-13PTD* parasites reveals the importance of kinesin-13 in regulating MT organisation in sexual stages in the mosquito. Kinesin-13s are MT depolymerising kinesins, playing essential roles in spindle MT dynamics, kinetochore-MT attachment, and chromosome segregation [36–38]. *Plasmodium falciparum* kinesin-13 has also been shown to exhibit MT depolymerisation activity in vitro [56]. Kinesin-13 homologues are present in most eukaryotes possessing cilia or flagella [57] and regulate the length of these structures [58–60]. The knockdown of *P. berghei* kinesin-13 resulted in defects in the organisation of both spindle and axonemal MTs, thus arresting nuclear division and axoneme assembly during male gametogenesis. A similar phenotype was observed for the kinesin-8B gene

knockout, which shows defective basal body formation and axoneme assembly during male gametogenesis, but nuclear division was normal [20]. Knockout of the gene for another basal body protein, SAS6, resulted in a similar phenotype with defective basal body formation and axoneme assembly but no effect on nuclear division [61]. Disruption of the gene for PF16, an armadillo-repeat protein of *Plasmodium* spp., produces a similar phenotype, with an effect on axonemal central MT assembly and male gamete motility and fertility [62]. We found a cdc2-related kinase (CRK5) that is important for nuclear spindle formation but has no effect on axoneme assembly during male gametogenesis [63]. Previous studies have shown a similar phenotype for CDPK4 and MAP2 gene disruptions, where either the early or late stages of exflagellation are impaired but axoneme assembly is not [22,64–66]. In another primitive eukaryote, *Giardia intestinalis*, kinesin-13 regulates MT dynamics during both flagellar assembly and nuclear division [67].

Kinesin-13 has an additional role during zygote to ookinete transformation as shown by kinesin-13PTD parasites producing retorts. Ultrastructure analysis of these retorts revealed a loss of polarity and disorganisation of the subpellicular MTs, consistent with the additional polar localization of kinesin-13GFP in zygotes and during ookinete development. This phenotype of kinesin-13 depletion is different from that of the *Δkinesin-20*, where apical polarity was not affected. A similar phenotype has been observed following knockdown of 2 *P. berghei* phosphatases, PPKL and PP1, where apical polarity was lost, affecting ookinete differentiation [49,68].

Global transcriptomic analysis supports the idea of a defect in spindle formation and axoneme assembly during male gametogenesis in kinesin-13 depleted parasites, as the expression of several genes involved in chromosome segregation, axoneme biogenesis, IMC/glideosome formation, and other biological processes were modulated in *kinesin-13PTD* parasites. For example, differentially expressed genes like CRK5, SRPK, and CDPK4 are involved in mitotic spindle formation during male gametogenesis [22,64,65]. Similarly, kinesin-8B, kinesin-X4, dynein, and radial spoke proteins are involved in axoneme assembly, male gamete formation, and fertility [20,43,69]. The compensatory expression of these genes is high in kinesin-13-depleted gametocytes, suggesting their role in the same pathway during male gametogenesis, but it is insufficient to rescue the phenotype. Differential expression of genes like IMCs and GAPs indicates the additional role of kinesin-13 in glideosome formation and motility. Most of these changes in gene expression are obvious at 15 mpa of gametocytes prior to gamete formation and fertilisation. Translation repression in *Plasmodium* spp. is released after fertilisation, allowing the stored transcripts in female gametocytes to be translated to form proteins essential for zygote development and ookinete invasion [30]. Differential expression, mainly up-regulation of these genes in *kinesin-13PTD* parasites, suggests a compensatory role during male gametogenesis and ookinete formation. Overall, these global transcriptomic data for *kinesin-13PTD* parasites are consistent with the profound phenotypic changes observed during male gametogenesis and ookinete formation.

In conclusion, the 9 *P. berghei* kinesins show a diverse pattern of expression and subcellular location at various stages of the parasite life cycle. Genetic and phenotypic analyses indicate that most kinesins have their most important roles in mosquito stages, except kinesin-13, which is also essential for asexual blood stages. Kinesin-20 and kinesin-13 have roles in MT dynamics during proliferation, polarity formation, and transmission of the parasite. It will be interesting in the future to look for the interacting partners of these kinesins and understand their mechanisms of action during these biological processes. This comprehensive study provides knowledge and understanding of the important roles of kinesins in various cellular processes at different stages of the life cycle of this evolutionarily divergent eukaryotic pathogen.

This information may be useful to exploit kinesins as potential targets for new therapeutic interventions against malaria.

## Materials and methods

### Ethics statement

The animal work passed an ethical review process and was approved by the United Kingdom Home Office. Work was carried out under UK Home Office Project Licences (30/3248 and PDD2D5182) in accordance with the UK "Animals (Scientific Procedures) Act 1986". Six- to 8-week-old female CD1 outbred mice from Charles River laboratories were used for all experiments.

### Generation of transgenic parasites and genotype analyses

To observe the location of kinesin proteins, the C-terminus was tagged with green fluorescent protein (GFP) sequence by single crossover homologous recombination at the 3′ end of the gene. To generate the GFP-tag line, a region of these genes downstream of the ATG start codon was amplified, ligated to p277 vector, and transfected as described previously [49]. The p277 vector contains the human *dhfr* cassette, conveying resistance to pyrimethamine. A schematic representation of the endogenous gene locus, the constructs, and the recombined gene locus can be found in **S1A Fig.** For the parasites expressing a C-terminal GFP-tagged protein, diagnostic PCR was used with primer 1 (Int primer) and primer 3 (ol492) to confirm integration of the GFP targeting construct (**S1B Fig**). A list of primers used to amplify these genes can be found in **S2 Table**.

The gene-deletion targeting vectors for *kinesin* genes were constructed using the pBS-DHFR plasmid, which contains polylinker sites flanking a *Toxoplasma gondii dhfr/ts* expression cassette conferring resistance to pyrimethamine, as described previously [22]. The 5′ upstream sequences from genomic DNA of kinesin genes were amplified and inserted into *Apa*I and *Hin*dIII restriction sites upstream of the *dhfr/ts* cassette of pBS-DHFR. The DNA fragments amplified from the 3′ flanking region of *kinesin* genes were then inserted downstream of the *dhfr/ts* cassette using *Eco*RI and *Xba*I restriction sites. The linear targeting sequence was released using *Apa*I/*Xba*I. A schematic representation of the endogenous *kinesin* loci, the constructs, and the recombined *kinesin* loci can be found in **S3 Fig**. The primers used to generate the mutant parasite lines can be found in **S2 Table**. A diagnostic PCR was used with primer 1 (Int primer) and primer 2 (ol248) to confirm integration of the targeting construct, and primer 3 (KO1) and primer 4 (KO2) were used to confirm deletion of the *kinesin* genes (**S3 Fig** and **S2 Table**).

To study the function of kinesin-13, we used 2 conditional knockdown systems; an AID (kinesin13AID) system and a promoter exchange/trap using double homologous recombination (kinesin-13PTD). For the generation of transgenic kinesin-13AID/HA line, library clone PbG01-2471h08 from the PlasmoGEM repository (http://plasmogem.sanger.ac.uk/) was used. Sequential recombineering and gateway steps were performed as previously described [70,71]. Insertion of the GW cassette following gateway reaction was confirmed using primer pairs GW1 x *kinesin-13* QCR1 and GW2 x *kinesin-13* QCR2. The modified library inserts were then released from the plasmid backbone using NotI. The kinesin-13-AID/HA targeting vector was transfected into the 615-parasite line and conditional degradation of kinesin-13-AID/HA in the nonclonal line was performed as described previously [63]. A schematic representation of the endogenous *kinesin-13* locus, the constructs, and the recombined *kinesin-13* locus can be found in **S7A Fig.** A diagnostic PCR was performed for *kinesin-13* gene knockdown parasites as outlined in **S7A Fig.** Primer pairs *Kinesin-13*QCR1/GW1 and *Kinesin-13* QCR2/GW2 were

used to determine successful integration of the targeting construct at the 3′ end of the gene (**S7B Fig**).

The conditional knockdown construct kinesin-13PTD was derived from $P_{clag}$ (pSS367) where *kinesin-13* was placed under the control of the *clag* gene (PBANKA_083630) promoter, as described previously [33]. A schematic representation of the endogenous *kinesin-13* locus, the constructs, and the recombined *kinesin-13* locus can be found in **S7D Fig**. A diagnostic PCR was performed for *kinesin-13* gene knockdown parasites as outlined in **S7D Fig**. Primer 1 (5′-intPTD50) and Primer 2 (5′-intPTD) were used to determine successful integration of the targeting construct at the 5′ end of the gene. Primer 3 (3′-intPTclag) and Primer 4 (3′-intPTD50) were used to determine successful integration for the 3′ end of the gene locus (**S7E Fig**). All the primer sequences can be found in **S2 Table**. *P. berghei* ANKA line 2.34 (for GFP-tagging) or ANKA line 507cl1 expressing GFP (for the gene deletion and knockdown construct) parasites were transfected by electroporation [72].

## Parasite phenotype analyses

Blood containing approximately 50,000 parasites of the kinesin knockout/knockdown lines was injected intraperitoneally (IP) into mice to initiate infection. Asexual stages and gametocyte production were monitored by microscopy on Giemsa-stained thin smears. Four to 5 dpi, exflagellation and ookinete conversion were examined as described previously [49] with a Zeiss AxioImager M2 microscope (Carl Zeiss) fitted with an AxioCam ICc1 digital camera. To analyse mosquito transmission, 30 to 50 *Anopheles stephensi* SD 500 mosquitoes were allowed to feed for 20 min on anaesthetized, infected mice whose asexual parasitaemia had reached 15% and were carrying comparable numbers of gametocytes as determined on Giemsa-stained blood films. To assess midgut infection, approximately 15 guts were dissected from mosquitoes on day 7 and 14 post-feeding and oocysts were counted using a 63× oil immersion objective. On day 21 post-feeding, another 20 mosquitoes were dissected, and their guts and salivary glands crushed separately in a loosely fitting homogeniser to release sporozoites, which were then quantified using a haemocytometer or used for imaging. Mosquito bite back experiments were performed 21 days post-feeding using naive mice, and blood smears were examined after 3 to 4 days.

## Purification of schizonts, gametocytes, and ookinetes

Blood cells obtained from infected mice (day 4 post-infection) were cultured for 8 h and 24 h at 37˚C (with rotation at 100 rpm), and schizonts were purified on a 60% v/v NycoDenz (in PBS) gradient, (NycoDenz stock solution: 27.6% w/v NycoDenz in 5 mM Tris–HCl (pH 7.20), 3 mM KCl, 0.3 mM EDTA).

The purification of gametocytes was achieved by injecting parasites into phenylhydrazine-treated mice [73] and enriched by sulfadiazine treatment after 2 days of infection. The blood was collected on day 4 after infection, and gametocyte-infected cells were purified on a 48% v/v NycoDenz (in PBS) gradient (NycoDenz stock solution: 27.6% w/v NycoDenz in 5 mM Tris–HCl (pH 7.20), 3 mM KCl, 0.3 mM EDTA). The gametocytes were harvested from the interface and activated.

Blood cells obtained from infected mice (day 4 to 5 post-infection) with high gametocytemia (>20%) were cultured for 24 h in ookinete medium containing xanthurenic acid at 20˚C. The ookinetes were pelleted at 1,900 rpm (500*g*), the supernatant was discarded, and the cells were resuspended in 8 ml ookinete medium. A volume of 5 μl of magnetic beads coated with 13.1 antibody (anti-P28 of ookinete) were added and incubated for 10 min at room temperature (RT) with continuous mixing. The tubes were placed on the magnet (Dyna rack) for 2

min, and supernatant was transferred into new tubes. The beads with bound ookinetes were washed with 2 ml of ookinete medium and used for imaging or electron microscopy.

## Immunoprecipitation and mass spectrometry

Male gametocytes 1 to 2 mpa of kinesin-GFP parasites were used to prepare cell lysates. WT-ANKA and non-kinesin proteins tagged with GFP were used as negative controls. Purified parasite pellets were crosslinked using formaldehyde (10 min incubation with 1% formaldehyde in PBS), followed by 5 min incubation in 0.125 M glycine solution and 3 washes with phosphate buffered saline (PBS, pH 7.5). Immunoprecipitation was performed using crosslinked protein lysate and a GFP-Trap_A Kit (Chromotek) following the manufacturer's instructions. Proteins bound to the GFP-Trap_A beads were digested using trypsin, and the peptides were analysed by liquid chromatography–tandem mass spectrometry. Mascot (http://www.matrixscience.com/) and MaxQuant (https://www.maxquant.org/) search engines were used for mass spectrometry data analysis. The PlasmoDB database was used for protein annotation. Peptide and proteins having minimum threshold of 95% were used for further proteomic analysis.

## Live cell imaging

To examine kinesin-GFP expression during erythrocyte stages, parasites growing in schizont culture medium were used for imaging at different stages (ring, trophozoite, schizont, and merozoite) of development. Purified gametocytes were examined for GFP expression and cellular location at different time points (0, 1 to 15 min) after activation in ookinete medium [18]. Zygote and ookinete stages were analysed throughout 24 h of culture. Oocysts and sporozoites were imaged using infected mosquito guts. Images were captured using a 63× oil immersion objective on a Zeiss Axio Imager M2 microscope fitted with an AxioCam ICc1 digital camera.

## Generation of dual tagged parasite lines

The kinesin-GFP (kinesin-5, kinesin-8X, kinesin-13, and kinesin-X4) parasites were mixed with NDC80-cherry and kinesin8B-cherry parasites in equal numbers and injected into mice. Mosquitoes were fed on mice 4 to 5 days after infection when gametocytemia was high. These mosquitoes were checked for oocyst development and sporozoite formation at day 14 and day 21 after feeding. Infected mosquitoes were then allowed to feed on naïve mice and after 4 to 5 days the mice were examined for blood stage parasitaemia by microscopy with Giemsa-stained blood smears. In this way, some parasites expressed both kinesin-GFP and NDC80-cherry, and kinesin-GFP and kinesin-8B-cherry in the resultant gametocytes, and these were purified and fluorescence microscopy images were collected as described above.

## Ookinete and sporozoite motility assays

Sporozoites were isolated from the salivary glands of mosquitoes infected with WTGFP and Δkinesin-20 parasites 21 dpi. Isolated sporozoites in RPMI 1640 containing 3% bovine serum albumin (Fisher Scientific) were pelleted (5 min, 5,000 rpm, 4˚C) and used for motility assay. The assay using Matrigel was performed as described previously [19,28]. A small volume (20 μl) of sporozoites, isolated as above for WTGFP and Δkinesin-20 parasites, were mixed with Matrigel (Corning, NY, USA). The mixture (6 μl) was transferred onto a microscope slide with a cover slip and sealed with nail polish. After identifying a field containing sporozoites, time-lapse videos (1 frame every 2 s for 100 cycles) were taken using the differential

interference contrast settings with a 63× objective lens on a Zeiss AxioImager M2 microscope fitted with an AxioCam ICc1 digital camera and analysed with the AxioVision 4.8.2 software.

For ookinete motility, 24-h ookinete cultures were added to an equal volume of Matrigel on ice, mixed thoroughly, dropped onto a slide, covered with a cover slip, and sealed with nail polish. The Matrigel was then allowed to set at 20°C for 30 min. After identifying a field containing ookinetes, time-lapse videos were taken (1 frame every 5 s for 100 cycles).

## Fixed immunofluorescence assay

The kinesin-GFP gametocytes were purified, activated in ookinete medium, fixed with 4% paraformaldehyde (PFA, Sigma) diluted in MT stabilising buffer (MTSB) for 10 to 15 min and added to poly-L-lysine coated slides. Immunocytochemistry was performed using primary GFP-specific rabbit mAb (Invitrogen-A1122; used at 1:250) and primary mouse anti-α tubulin mAb (Sigma-T9026; used at 1:1,000). Secondary antibodies were Alexa 488 conjugated anti-mouse IgG (Invitrogen-A11004) and Alexa 568–conjugated anti-rabbit IgG (Invitrogen-A11034) (used at 1 in 1,000). The slides were then mounted in Vectashield 19 with DAPI (Vector Labs) for fluorescence microscopy. Parasites were visualised on a Zeiss AxioImager M2 microscope fitted with an AxioCam ICc1 digital camera.

## Ultrastructure expansion microscopy (U-ExM)

Purified gametocytes were activated for 4 to 5 min and 15 min; activation was stopped by adding 1X ice-cold PBS. Activated gametocytes and mature ookinetes were sedimented onto 12-mm round Poly-D-Lysine (A3890401, Gibco) coated coverslips for 10 min (gametocyte procedure was performed on ice), fixed in methanol at −20°C for 7 min, and then prepared for U-ExM as described previously [45,74]. Immuno-labelling was performed using primary antibody against α-tubulin and β-tubulin (1:200 dilution, source: Geneva antibody facility) and secondary antibody anti-guinea pig Alexa 488 (1:400 dilution, source: ThermoFisher). Images were acquired on a Leica TCS SP8 microscope; image analysis was performed using Fiji-Image J and Leica Application Suite X (LAS X) software.

## Electron microscopy

Gametocytes activated for 6 min and 15 min, and ookinetes were fixed in 4% glutaraldehyde in 0.1 M phosphate buffer and processed for electron microscopy [75]. Briefly, samples were post-fixed in osmium tetroxide, treated en bloc with uranyl acetate, dehydrated and embedded in Spurr's epoxy resin. Thin sections were stained with uranyl acetate and lead citrate prior to examination in a JEOL JEM-1400 electron microscope (JEOL, UK).

## RNA isolation and quantitative real-time PCR (qRT-PCR) analyses

RNA was isolated from purified gametocytes using an RNA purification kit (Stratagene). cDNA was synthesised using an RNA-to-cDNA kit (Applied Biosystems). Gene expression was quantified from 80 ng of total RNA using SYBR green fast master mix kit (Applied Biosystems). All the primers were designed using primer3 (Primer-blast, NCBI). Analysis was conducted using an Applied Biosystems 7500 fast machine with the following cycling conditions: 95°C for 20 s followed by 40 cycles of 95°C for 3 s; 60°C for 30 s. Three technical replicates and 3 biological replicates were performed for each assayed gene. The *hsp70* (PBANKA_081890) and *arginyl-t RNA synthetase* (PBANKA_143420) genes were used as endogenous control reference genes. The primers used for qPCR can be found in **S2 Table**.

## RNA-seq analysis

Libraries were prepared from lyophilized total RNA, first by isolating mRNA using NEBNext Poly(A) mRNA Magnetic Isolation Module (NEB), then using NEBNext Ultra Directional RNA Library Prep Kit (NEB, MA, USA) according to the manufacturer's instructions. Libraries were amplified for a total of 12 PCR cycles (12 cycles of [15 s at 98°C, 30 s at 55°C, 30 s at 62°C]) using the KAPA HiFi HotStart Ready Mix (KAPA Biosystems, MA, USA). Libraries were sequenced using a NovaSeq 6000 DNA sequencer (Illumina, CA, USA), producing paired-end 100-bp reads.

FastQC (https://www.bioinformatics.babraham.ac.uk/projects/fastqc/) was used to analyse raw read quality, and based on this information, the first 11 bp of each read and any adapter sequences were removed using Trimmomatic (http://www.usadellab.org/cms/?page=trimmomatic). Bases were trimmed from reads using Sickle with a Phred quality threshold of 25 (https://github.com/najoshi/sickle). The resulting reads were mapped against the *P. berghei* ANKA genome (v36) using HISAT2 (version 2–2.1.0), using default parameters. Uniquely mapped, properly paired reads with mapping quality 40 or higher were retained using SAMtools (http://samtools.sourceforge.net/). Genome browser tracks were generated and viewed using the Integrative Genomic Viewer (IGV) (Broad Institute). Raw read counts were determined for each gene in the *P. berghei* genome using BedTools (https://bedtools.readthedocs.io/en/latest/#) to intersect the aligned reads with the genome annotation. Differential expression analysis was done by use of R package DESeq2 to call up- and down-regulated genes with an adjusted *P* value cutoff of 0.05. Gene ontology enrichment was done using R package topGO (https://bioconductor.org/packages/release/bioc/html/topGO.html) with the weight01 algorithm.

## ChIP-seq analysis

Gametocytes for kinesin-5GFP, kinesin-8XGFP, and NDC80GFP (as a positive control) parasites were harvested, and the pellets were resuspended in 500 μl of Hi-C lysis buffer (25 mM Tris–HCl (pH 8.0), 10 mM NaCl, 2 mM AESBF, 1% NP-40, protease inhibitors). After incubation for 10 min at RT, the resuspended pellets were homogenised by passing through a 26.5 gauge needle/syringe 15 times and cross-linked by adding formaldehyde (1.25% final concentration) for 25 min at RT with continuous mixing. Crosslinking was stopped by adding glycine to a final concentration of 150 mM and incubating for 15 min at RT with continuous mixing. The sample was centrifuged for 5 min at 2,500 × *g* (approximately 5,000 rpm) at 4°C, the pellet washed once with 500 μl ice-cold wash buffer (50 mM Tris–HCl (pH 8.0), 50 mM NaCl, 1 mM EDTA, 2 mM AESBF, protease inhibitors) and the pellet stored at −80°C for ChIP-seq analysis. The crosslinked parasite pellets were resuspended in 1 mL of nuclear extraction buffer (10 mM HEPES, 10 mM KCl, 0.1 mM EDTA, 0.1 mM EGTA, 1 mM DTT, 0.5 mM AEBSF, 1X protease inhibitor tablet), post 30-min incubation on ice, 0.25% Igepal-CA-630 was added and homogenised by passing through a 26G × ½ needle. The nuclear pellet extracted through 5,000 rpm centrifugation was resuspended in 130 μl of shearing buffer (0.1% SDS, 1 mM EDTA, 10 mM Tris–HCl (pH 7.5), 1X protease inhibitor tablet) and transferred to a 130-μl Covaris sonication microtube. The sample was then sonicated using a Covaris S220 Ultrasonicator for 8 min (duty cycle: 5%, intensity peak power: 140, cycles per burst: 200, bath temperature: 6°C). The sample was transferred to ChIP dilution buffer (30 mM Tris–HCl (pH 8), 3 mM EDTA, 0.1% SDS, 30 mM NaCl, 1.8% Triton X-100, 1X protease inhibitor tablet, 1X phosphatase inhibitor tablet) and centrifuged for 10 min at 13,000 rpm at 4°C, retaining the supernatant. For each sample, 13 μl of protein A agarose/salmon sperm DNA beads were washed 3 times with 500 μl ChIP dilution buffer (without inhibitors) by centrifuging for 1 min at 1,000 rpm at RT, then buffer was removed. For preclearing, the diluted chromatin samples were added to the beads

and incubated for 1 h at 4°C with rotation, then pelleted by centrifugation for 1 min at 1,000 rpm. Before adding antibody, approximately 10% of 1 kin-8XGFP sample was taken as input. Supernatant was removed into a LoBind tube carefully so as not to remove any beads, and 2 μg of anti-GFP antibody (Abcam ab290, anti-rabbit) were added to the sample and incubated overnight at 4°C with rotation. For 1 kinesin-8XGFP sample, IgG antibody (ab46540) was added instead as a negative control. Per sample, 25 μl of protein A agarose/salmon sperm DNA beads were washed with ChIP dilution buffer (no inhibitors), blocked with 1 mg/mL BSA for 1 h at 4°C, then washed 3 more times with buffer. A volume of 25 μl of washed and blocked beads were added to the sample and incubated for 1 h at 4°C with continuous mixing to collect the antibody/protein complex. Beads were pelleted by centrifugation for 1 min at 1,000 rpm at 4°C. The bead/antibody/protein complex was then washed with rotation using 1 mL of each buffers twice; low-salt immune complex wash buffer (1% SDS, 1% Triton X-100, 2 mM EDTA, 20 mM Tris–HCl (pH 8), 150 mM NaCl), high-salt immune complex wash buffer (1% SDS, 1% Triton X-100, 2 mM EDTA, 20 mM Tris–HCl (pH 8), 500 mM NaCl), high-salt immune complex wash buffer (1% SDS, 1% Triton X-100, 2 mM EDTA, 20 mM Tris–HCl (pH 8), 500 mM NaCl), TE wash buffer (10 mM Tris–HCl (pH 8), 1 mM EDTA) and eluted from antibody by adding 250 μl of freshly prepared elution buffer (1% SDS, 0.1 M sodium bicarbonate). We added 5 M NaCl to the elution and cross-linking was reversed by heating at 45°C overnight followed by addition of 15 μl of 20 mg/mL RNAase A with 30-min incubation at 37°C. After this, 10 μl 0.5 M EDTA, 20 μl 1 M Tris–HCl (pH 7.5), and 2 μl 20 mg/mL proteinase K were added to the elution and incubated for 2 h at 45°C. DNA was recovered by phenol/chloroform extraction and ethanol precipitation, using a phenol/chloroform/isoamyl alcohol (25:24:1) mixture twice and chloroform once, then adding 1/10 volume of 3 M sodium acetate (pH 5.2), 2 volumes of 100% ethanol, and 1/1,000 volume of 20 mg/mL glycogen. Precipitation was allowed to occur overnight at −20°C. Samples were centrifuged at 13,000 rpm for 30 min at 4°C, then washed with fresh 80% ethanol, and centrifuged again for 15 min with the same settings. Pellet was air dried and resuspended in 50 μl nuclease-free water. DNA was purified using Agencourt AMPure XP beads (Beckman Coulter, CA, USA). Libraries were then prepared from this DNA using a KAPA library preparation kit (KK8230) and sequenced on a NovaSeq 6000 machine. FastQC (https://www.bioinformatics.babraham.ac.uk/projects/fastqc/), was used to analyse raw read quality. Any adapter sequences were removed using Trimmomatic (http://www.usadellab.org/cms/?page=trimmomatic). Bases with Phred quality scores below 25 were trimmed using Sickle (https://github.com/najoshi/sickle). The resulting reads were mapped against the *P. berghei* ANKA genome (v36) using Bowtie2 (version 2.3.4.1). Using Samtools, only properly paired reads with mapping quality 40 or higher were retained and reads marked as PCR duplicates were removed by PicardTools MarkDuplicates (Broad Institute). Genome-wide read counts per nucleotide were normalised by dividing millions of mapped reads for each sample (for all samples including input) and subtracting input read counts from the ChIP and IgG counts. From these normalised counts, genome browser tracks were generated and viewed using the IGV.

## Statistical analysis

All statistical analyses were performed using GraphPad Prism 7 (GraphPad Software). For qRT-PCR, an unpaired *t* test was used to examine significant differences between wild-type and mutant strains.

## Supporting information

**S1 Fig. Generation and genotypic analysis of kinesin-GFP parasites. (A)** Schematic representation of the endogenous *kinesin* locus, the GFP-tagging construct, and the recombined

*kinesin* locus following single homologous recombination. Arrows 1 (P1) and 3 (P3) indicate the position of PCR primers used to confirm successful integration of the construct. (**B**) Diagnostic PCR of *kinesin* and WT parasites using primers: integration primer (P1) and ol492 (P2). The bands of expected size for amplified DNA fragments are indicated for each tagged line.
(TIF)

**S2 Fig. Immunoprecipitation and mass-spectrometric analysis of kinesins.** (**A**) The number of peptides after immunoprecipitation using GFP-trap beads and their coverage to full length kinesins. (**B**) Locations of peptides in full-length kinesin proteins.
(TIF)

**S3 Fig. Kinesin pattern of expression and diverse subcellular locations at various stages in the *Plasmodium berghei* life cycle.** Live cell imaging showing subcellular locations of 9 kinesin-GFP proteins (green) during various stages of the *P. berghei* life cycle. DNA is stained with Hoechst dye (blue). Arrowhead indicates basal end and arrow indicates apical end of the ookinete. Scale bar = 5 μm. mpa, min post-activation; Oocyst-NS, nonsporulating oocyst; Oocyst-S, sporulating oocyst.
(TIF)

**S4 Fig. Generation and genotypic analysis of *Δkinesin* parasites.** (**A**) Schematic representation of the endogenous *kinesin* locus, the targeting knockout construct, and the recombined *kinesin* locus following double homologous crossover recombination. Arrows 1 and 2 indicate PCR primers used to confirm successful integration in the *kinesin* locus following recombination, and arrows 3 and 4 indicate PCR primers used to show deletion of the *kinesin* gene. (**B**) Integration PCR of the *kinesin* locus in WTGFP (WT) and knockout (Mut) parasites using primers: integration primer and ol248. Integration of the targeting construct gives expected size band for each gene.
(TIF)

**S5 Fig. Phenotypic screen of 9 kinesins in *Plasmodium* reveals their role during sexual and transmission stages within mosquito vector.** (**A**) Male gametogenesis in kinesin gene-deletion lines in comparison with WTGFP parasite, measured as the number of exflagellation centres per field; more than 20 fields were counted for each line. Mean ± SEM. $n = 3$ independent experiments. (**B**) Percentage ookinete conversion comparing knockout and WTGFP parasites. Ookinetes were identified using 13.1 antibody for surface marker (P28, red) and defined as those cells that differentiated successfully into elongated "banana-shaped" ookinetes. Mean ± SEM. $n = 5$ independent experiments. (**C**) Total number of GFP-positive oocysts per mosquito midgut at 7, 14, and 21 dpi for knockout and WTGFP parasites; at least 10 mosquito midguts were counted for each line. Mean ± SEM. $n = 3$ independent experiments. (**D**) Representative mosquito midguts at 10× and 63× magnification showing oocysts of kinesin knockout and WTGFP lines at 7, 14, and 21 dpi. Scale bar = 50 μm (10×), 20 μm (63×). (**E**) Total number of sporozoites in oocysts of kinesin knockout and WTGFP parasites at 14 and 21 dpi. Mean ± SEM. $n = 3$ independent experiments. (**F**) Total number of sporozoites in salivary glands of kinesin knockout and WT-GFP parasites. Mean ± SEM. $n = 3$ independent experiments. (**G**) Mosquito bite back experiments showing no transmission of *Δkinesin-8B* and *Δkinesin-8X* parasites, while other kinesin knockout and WTGFP parasites show successful transmission from mosquito to mice. Mean ± SEM. $n = 3$ independent experiments. $^*p \leq 0.05$, $^{**}p \leq 0.01$, and $^{***}p \leq 0.001$. Underlying data are provided in the Supporting information as S3 Data. dpi, days post-infection.
(TIF)

**S6 Fig. The location of kinesin-5 and kinesin-8X in relation to that of kinetochore (NDC80) and axoneme marker (kinesin-8B). (A)** Live cell images of gametocytes showing dual lines expressing either kinesin-5GFP or NDC80-mCherry/kinesin-8BmCherry or both in the same gametocyte. **(B)** Live cell imaging showing the temporal dynamics of kinesin-5GFP (green) along with kinetochore marker NDC80Cherry (red) during male gametogenesis. DNA is stained with Hoechst dye (blue); scale bar = 5 μm. **(C)**. The location of kinesin-5GFP (green) in relation to the axoneme marker, kinesin-8BCherry (red) during male gamete formation. **(D)**. The location of kinesin-8XGFP (green) in relation to the axoneme marker, kinesin-8BCherry (red) during male gamete formation. The nuclear location of kinesin-5 and kinesin-8X contrasts with the cytoplasmic location of kinesin-8B during chromosome replication and segregation, indicating that kinesin-5 and kinesin-8X are associated with the mitotic spindle. DNA is stained with Hoechst dye (blue). Scale bar = 5 μm. mpa, min post-activation; (TIF)

**S7 Fig. ChIP-seq analysis of kinesin-8X and kinesin-5 during gametocyte stage.** Centromeric localization confirmed by ChIP-seq analysis of kinesin-8XGFP and kinesin-5GFP profiles for all 14 chromosomes in gametocyte stage. Signals are plotted on a normalised RPM basis. Lines on top are division points between chromosomes, and circles on the bottom indicate locations of centromeres. NDC80-GFP was used as a positive control, and IgG was used as a negative control. IgG, immunoglobulin G; RPM, read per million. (TIF)

**S8 Fig. qRT-PCR and RNA-seq analysis and sporozoite motility of *Δkinesin-20* parasite lines. (A)** Representative frames from time-lapse videos showing motile sporozoites of WTGFP and *Δkinesin-20* lines. Black arrow indicates the apical end of the sporozoites. Scale bar = 5 μm. **(B)** Sporozoite motility for WTGFP and *Δkinesin-20* lines. More than 20 sporozoites were examined for each line. Mean ± SEM. $n$ = 3 independent experiments. **(C)** qRT-PCR analysis of transcripts for other kinesin genes in *Δkinesin-20* and WTGFP parasites. Mean ± SEM. $n$ = 3 independent experiments. **(D)** RNA-seq analysis showing no transcript in *Δkinesin-20* parasites. **(E)** Volcano plot showing differentially regulated genes in *Δkinesin-20* gametocytes activated for 2 h. Underlying data are provided in the Supporting information as S4 Data. ns, not significant; qRT-PCR, quantitative real-time PCR. (TIF)

**S9 Fig. Generation and genotype analysis of conditional knockdown *kinesin-13* parasites. (A)** Schematic representation of AID strategy to generate *kinesin-13AID/HA* parasites. **(B)** Integration PCR of the kinesin-13AID/HA construct in the *kinesin-13* locus. Oligonucleotides used for PCR genotyping are indicated and agarose gels for corresponding PCR products from genotyping reactions are shown. **(C)** Kinesin-13AID/HA protein expression level as measured by western blotting upon addition of auxin to mature purified gametocytes; α-tubulin serves as a loading control. **(D)** Schematic representation of the promoter swap strategy (*kinesin-13PTD*, placing *kinesin-13* under the control of the *clag* promoter) by double homologous recombination. Arrows 1 and 2 indicate the primer positions used to confirm 5′ integration, and arrows 3 and 4 indicate the primers used for 3′ integration. **(E)** Integration PCR of the promotor swap construct into the *kinesin-13* locus. Primer 1 (5′-IntPTD50) with primer 2 (5′-IntPTD) were used to determine successful integration of the selectable marker. Primer 3 (3′-intClag) and primer 4 (3′-IntPTD50) were used to determine the successful integration of the *clag* promoter. Primer 1 (5′-IntPTD50) and primer 4 (3′-IntPTD50) were used to show complete knock-in of the construct and the absence of a band at 2.3 kb (endogenous) expected if no integration occurred. **(F)** Oocysts at days 7, 14, and 21 post-infection. $n$ = 3 independent

experiments with a minimum of 8 mosquito guts. Error bar, ±SEM. (**G**) Bite back experiments reveal no transmission of *kinesin-13PTD* and successful transmission of WTGFP parasites from mosquito to mouse. Mean ± SEM. *n* = 3 independent experiments. Underlying data are provided in the Supporting information as S5 Data. AID, auxin-inducible degron.
(TIF)

**S1 Table. List of genes differentially expressed between Δ*kinesin-20* and WTGFP activated gametocytes for 2 h (S1A Table) and between *kinesin-13PTD* and WTGFP gametocytes (S1B and S1C Table).**
(XLS)

**S2 Table. Oligonucleotides used in this study.**
(XLS)

**S1 Movie. Gliding motility of WT-GFP ookinetes.**
(AVI)

**S2 Movie. Gliding motility of Δ*kinesin-20* ookinetes.**
(AVI)

**S3 Movie. Gliding motility of WT-GFP salivary gland sporozoite.**
(AVI)

**S4 Movie. Gliding motility Δ*kinesin-20* salivary gland sporozoite.**
(AVI)

**S1 Data. Excel spreadsheet containing the underlying numerical data for Fig 3C, 3G and 3E.**
(XLSX)

**S2 Data. Excel spreadsheet containing the underlying numerical data for Fig 6C–6F.**
(XLSX)

**S3 Data. Excel spreadsheet containing the underlying numerical data for S5A–S5G Fig.**
(XLSX)

**S4 Data. Excel spreadsheet containing the underlying numerical data for S8B and S8C Fig.**
(XLSX)

**S5 Data. Excel spreadsheet containing the underlying numerical data for S9F and S9G Fig.**
(XLSX)

**S1 Raw Images. Original gel and blot images.**
(PDF)

## Acknowledgments

We wish to thank Julie Rodgers for helping to maintain the insectary and other technical works.

## Author Contributions

**Conceptualization:** Rita Tewari.

**Data curation:** Mohammad Zeeshan, Rita Tewari.

**Formal analysis:** Mohammad Zeeshan, Ravish Rashpa, David J. P. Ferguson, Steven Abel, Mathieu Brochet, Rita Tewari.

**Funding acquisition:** Karine G. Le Roch, Mathieu Brochet, Anthony A. Holder, Rita Tewari.

**Investigation:** Mohammad Zeeshan, Ravish Rashpa, David J. P. Ferguson, Steven Abel, Zeinab Chahine, Declan Brady, Rita Tewari.

**Methodology:** Mohammad Zeeshan, Ravish Rashpa, David J. P. Ferguson, Steven Abel, Zeinab Chahine, Declan Brady, Rita Tewari.

**Project administration:** Rita Tewari.

**Resources:** Sue Vaughan, Karine G. Le Roch, Mathieu Brochet, Rita Tewari.

**Software:** Sue Vaughan, Karine G. Le Roch.

**Supervision:** Anthony A. Holder, Rita Tewari.

**Validation:** Mohammad Zeeshan, David J. P. Ferguson, Steven Abel, Declan Brady, Rita Tewari.

**Visualization:** Mohammad Zeeshan, Ravish Rashpa, David J. P. Ferguson, Rita Tewari.

**Writing – original draft:** Mohammad Zeeshan, Rita Tewari.

**Writing – review & editing:** Mohammad Zeeshan, David J. P. Ferguson, Carolyn A. Moores, Karine G. Le Roch, Mathieu Brochet, Anthony A. Holder, Rita Tewari.

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
