## [Editor Report · Decision Letter 0]

30 Mar 2022

Dear Dr Tewari, 

Thank you for submitting your manuscript entitled "Key roles for kinesin-13 and kinesin-20 in malaria parasite proliferation, polarity, and transmission revealed by genome-wide functional analysis" for consideration as a Research Article by PLOS Biology.

Your manuscript has now been evaluated by the PLOS Biology editorial staff, as well as by an academic editor with relevant expertise, and I am writing to let you know that we would like to send your submission for re-review by the previous reviewers.

Once your full submission is complete, your paper will undergo a series of checks in preparation for peer review. Once your manuscript has passed the checks it will be sent out for review. To provide the metadata for your submission, please Login to Editorial Manager (https://www.editorialmanager.com/pbiology) within two working days, i.e. by Apr 01 2022 11:59PM.

Given the disruptions resulting from the ongoing COVID-19 pandemic, please expect some delays in the editorial process. We apologise in advance for any inconvenience caused and will do our best to minimize impact as far as possible.

Kind regards,

Dario

Dario Ummarino, PhD

Senior Editor

PLOS Biology

dummarino@plos.org

---

## [Decision Letter · Decision Letter 1]

6 May 2022

Dear Dr Tewari,

Thank you for your patience while we considered your revised manuscript "Key roles for kinesin-13 and kinesin-20 in malaria parasite proliferation, polarity, and transmission revealed by genome-wide functional analysis" for publication as a Research Article at PLOS Biology. This revised version of your manuscript has been evaluated by the PLOS Biology editors, the Academic Editor and one of the original reviewers. One of the reviewers who originally reviewed your manuscript at Review Commons (reviewer 3) was not available for re-review, therefore the Academic Editor assessed directly your responses to their original concerns. 

Based on the reviews and the Academic Editor's assessment of your revision, we are likely to accept this manuscript for publication, provided you satisfactorily address the remaining points raised by the reviewers and Academic Editors, which are pasted at the end of this email. Please also make sure to address the following data and other policy-related requests:

1) Title: We would like to suggest a modification for clarity and appeal: "Genome-wide functional analysis reveals key roles for kinesins in the mammalian and mosquito stages of the malaria parasite life cycle".

2) Blurb: Please provide a blurb which (if accepted) will be included in our weekly and monthly Electronic Table of Contents, sent out to readers of PLOS Biology, and may be used to promote your article in social media. The blurb should be about 30-40 words long and is subject to editorial changes. It should, without exaggeration, entice people to read your manuscript. It should not be redundant with the title and should not contain acronyms or abbreviations. For examples, view our author guidelines: https://journals.plos.org/plosbiology/s/revising-your-manuscript#loc-blurb.

3) Data: You may be aware of the PLOS Data Policy, which requires that all data be made available without restriction: http://journals.plos.org/plosbiology/s/data-availability. For more information, please also see this editorial: http://dx.doi.org/10.1371/journal.pbio.1001797.

Note that we do not require all raw data. Rather, we ask for all individual quantitative observations that underlie the data summarized in the figures and results of your paper. For an example see here: http://www.plosbiology.org/article/info%3Adoi%2F10.1371%2Fjournal.pbio.1001908#s5

These data can be made available in one of the following forms:

I) Supplementary files (e.g., excel). Please ensure that all data files are uploaded as 'Supporting Information' and are invariably referred to (in the manuscript, figure legends, and the Description field when uploading your files) using the following format verbatim: S1 Data, S2 Data, etc. Multiple panels of a single or even several figures can be included as multiple sheets in one excel file that is saved using exactly the following convention: S1_Data.xlsx (using an underscore).

II) Deposition in a publicly available repository. Please also provide the accession code or a reviewer link so that we may view your data before publication.

Regardless of the method selected, please ensure that you provide the individual numerical values that underlie the summary data displayed in the following figure panels: Figures 3 CGE, 6 CDEFHIJ, S5 ABCEFG, S9 FG.

3.1) IMPORTANT: Please also cite the location of the data clearly in each relevant main and supplementary Figure legend, e.g. “Data underlying this Figure can be found in S1 Data”.

3.2) Please ensure that your Data Statement in the submission system accurately describes where your data can be found.

4) Gel/Blot data: Please provide the original, uncropped and minimally adjusted images supporting the blot and gel results reported in Figures S1B, S4B, S9 BCE. Our guidelines for how to prepare and upload this data can be found here: https://journals.plos.org/plosbiology/s/figures#loc-blot-and-gel-reporting-requirements

We expect to receive your revised manuscript within two three weeks. 

*Published Peer Review History*

*Press*

Sincerely,

Dario

Dario Ummarino, PhD

Senior Editor

PLOS Biology

dummarino@plos.org

Reviewer's report:

Reviewer #1: The authors addressed all comments from the Review Commons published in December 2021. 

I have one last request upon endorsement for publication is to remove the term "endomitotic" in line 87.

Endomitosis means that the number of chromosomes increases without nuclear division, leading to the polyploid stage. In contrast, the Plasmodium parasite replicates by schizogony, where DNA replication is followed by nuclear division (PMID: 35353560). For instance, Sarcocystis replicates by endomitosis, not Plasmodium (suggested references are PMID: 17604449 and 33912479). 

Understanding the function of microtubule motors in Plasmodium spp will advance our knowledge of the atypical cell division mode of the malaria parasite and unveil innovative ways to block the parasite replication.

Comments from the Academic Editor:

Point 1. Details are missing in the methods for how the authors performed the immunoprecipitation with the GFP-TRAP beads and there is also no mention of what served as the negative control. Moreover, the peptide counts in the negative controls should also be included in S2 Fig so we can see what the background counts are like.

Also regarding GFP expression, Fig 1B and 1C don’t completely talk to one another. Because of the large number of proteins they are looking at, they show 1B for all the various localisations and then then 1C as a summative of what they saw (for which we don't see the raw data). In Fig 1B, The GFP signal either localises to the Spindle Pole or Spindle in male gametocytes and it would be good to know which kinesins these were. There are also no pictures of female gametocytes, male gametes or zygotes in Fig 1 B. Whilst it has been previously published that Kinesin 5 and 8X co-localise with anti-tubulin antibodies (to detect spindle), I think it is important to show a couple of co-localisation figures in this manuscript since the focus is on spindles and the reader should not have to go to another publication to see what that labelling looks like.

Point 2. Chip-Seq data is presented in S7Fig and includes positive and negative controls as requested. However, information regarding the controls should also be in the methodology section.

Point 3. Whilst the authors have now made a change in the text regarding why they undertook RNAseq analysis on the gam stage, they didn't actually point out that their transcriptomic data showed no role for kinesin-20 in translational repression. This should be specified as it is simply inferred but not overly clear. Also line349: comma should be replaced with full stop. This suggests a link…..

---

## [Editor Report · Decision Letter 2]

10 Jun 2022

Dear Dr Tewari,

Thank you for the submission of your revised Research Article "Genome-wide functional analysis reveals key roles for kinesins in the mammalian and mosquito stages of the malaria parasite life cycle" for publication in PLOS Biology. On behalf of my colleagues and the Academic Editor, Tania de Koning-Ward, I am pleased to say that we can in principle accept your manuscript for publication, provided you address any remaining formatting and reporting issues. These will be detailed in an email you should receive within 2-3 business days from our colleagues in the journal operations team; no action is required from you until then. Please note that we will not be able to formally accept your manuscript and schedule it for publication until you have completed any requested changes.

PRESS

Sincerely, 

Dario

Dario Ummarino, PhD

Senior Editor

PLOS Biology